# Early warning signals of the termination of the African Humid Period(s)

Martin H. Trauth [1] ✉, Asfawossen Asrat[2,3], Markus L. Fischer [1], Peter O. Hopcroft [4], Verena Foerster [5], Stefanie Kaboth-Bahr[6], Karin Kindermann[7], Henry F. Lamb [8,9], Norbert Marwan [10], Mark A. Maslin [11], Frank Schaebitz[5] & Paul J. Valdes [12]

The transition from a humid green Sahara to today's hyperarid conditions in northern Africa ~5.5 thousand years ago shows the dramatic environmental change to which human societies were exposed and had to adapt to. In this work, we show that in the 620,000-year environmental record from the Chew Bahir basin in the southern Ethiopian Rift, with its decadal resolution, this one thousand year long transition is particularly well documented, along with 20–80 year long droughts, recurring every ~160 years, as possible early warnings. Together with events of extreme wetness at the end of the transition, these droughts form a pronounced climate "flickering", which can be simulated in climate models and is also present in earlier climate transitions in the Chew Bahir environmental record, indicating that transitions with flickering are characteristic of this region.

The mid-Holocene climate transition from predominantly wet to dry conditions in tropical and subtropical northern Africa is the most dramatic example of a climate tipping point during the present interglacial. Climate tipping points occur when small perturbations in the forcing mechanism trigger a large, nonlinear response from the system, moving climate to a different future state generally accompanied by dramatic consequences for the biosphere[1–4]. The mid-Holocene climate transition underscores that the much-touted stability of the Holocene climate and its beneficial effects on the evolution of human societies do not hold true, at least not for the low latitudes. In fact, the transition from the African Humid Period (AHP), and with it the so-called Green Sahara phase, 15–5 kiloyears before present (kyr BP) to pronounced aridity after ~5 kyr BP led to significant changes in human environments in large areas of northern Africa[3,5–7].

The mid-Holocene climate transition was caused by the decrease in solar radiation in the Northern Hemisphere driven by the Earth's climate precession[3,8]. This change occurred slowly, as the sinusoidal period of precession is ~23 kyrs, whereas the more rapid quasilinear change in solar irradiance occurred during a quarter period of precession (~6 kyrs) between 9 and 3 kyr BP[9]. The response of the climate system to the 7–8% decrease in solar irradiance within 6 kyrs was much more rapid, perhaps faster than ~200 years in western Africa, but up to 1000 years in eastern Africa[3,7–9]. The main reason suggested for the rapid termination of the AHP is the positive feedback between the monsoon and vegetation that amplifies the comparatively small changes in external forcing[5].

These rapid environmental changes had a strong impact on humans in northern Africa, as their preferred habitats of grasslands, open forests, and lakes disappeared[10]. They responded to increasing

[1]University of Potsdam, Institute of Geosciences, Potsdam, Germany. [2]Botswana University of Science and Technology, Department of Mining and Geological Engineering, Palapye, Botswana. [3]Addis Ababa University, School of Earth Sciences, Addis Ababa, Ethiopia. [4]University of Birmingham, School of Geography, Earth & Environmental Sciences, Birmingham, United Kingdom. [5]University of Cologne, Institute of Geography Education, Cologne, Germany. [6]Freie Universität Berlin, Institute of Geological Sciences, Berlin, Germany. [7]University of Cologne, Institute of Prehistoric Archaeology, Cologne, Germany. [8]Aberystwyth University, Department of Geography and Earth Sciences, Aberystwyth, UK. [9]Trinity College Dublin, Botany Department, School of Natural Sciences, Dublin, Ireland. [10]Potsdam Institute for Climate Impact Research, Member of the Leibniz Association, Potsdam, Germany. [11]University College London, Geography Department, London, UK. [12]University of Bristol, Bristol Research Initiative for the Dynamic Global Environment, School of Geographical Sciences, Bristol, UK. ✉e-mail: trauth@uni-potsdam.de

aridity by retreating to regions with better water availability, such as mountain refuges, oases and the Nile valley[11,12]. Only a few settlements dated to the AHP have been recorded in the Nile valley so far[11]. Due to marshy areas with frequent, extensive flooding, wild animals (e.g., crocodiles) and disease-bearing mosquitos, the Nile valley may have been rather unattractive for settlement during this period[11–14]. With the onset of drier conditions at the end of the AHP, the Nile valley developed into a more suitable human habitat, with favorable conditions for farmers and livestock keepers, and for the subsequent development for a more complex society[11,12]. In this respect, the end of the AHP is an example of both the negative and positive effects of a climate tipping point on early societies[6,11].

The mid-Holocene climate tipping point in tropical and sub-tropical northern Africa has been the subject of much research[2–4]. This is because there is a comparatively simple but nonlinear relationship between the cause (orbital forcing) and the accelerated response of the monsoon system. It is also much studied because current human-induced climate change may reverse this climate tipping point; modeling results suggest that the green parts of the Sahel are spreading northward[3]. Recent literature distinguishes two major types of tipping points according to the nature of the cause and response of the climate system[3,15]. Tipping points of the first type are characterized by critical slowing down and a decreasing recovery from perturbations near the transition[16,17]. When the noise level is high, as in the African Monsoon System, however, we encounter a tipping point of a different type, namely a tipping accompanied by flickering between two stable states near the transition[2–4,17].

The two types of tipping points differ in the nature of the early warning signals, which are increasingly becoming the focus of research as they are particularly important to understand in order to predict possible future human-induced climate tipping points[16,17]. While slowing down in the first type of tipping point leads to a decrease in variability, autocorrelation and skewness, flickering in the second type leads to exactly the opposite, and, in case of doubt, to the impending tipping point not being recognized[16,17]. It is therefore important that the underlying processes are faithfully captured by models, decisively influencing the credibility of model-based predictions of abrupt changes under future climate forcing.

For the African Monsoon System, flickering prior to transition was recently predicted by a modeling study using HadCM3[15], whereas records of past climate changes are of too low resolution to comprehensively validate these model simulations[16,17]. These studies indicate a strong positive feedback between vegetation cover and precipitation, caused by both radiative and hydrological effects. First, the darker vegetation enhances solar energy absorbed fueling monsoonal type circulation onto land and hence precipitation, and second, vegetation as opposed to bare soil enables more efficient moisture-recycling back to the atmosphere, thus providing a pair of self-reinforcing feedbacks. In HadCM3, the albedo effect is stronger than the moisture effect by about a factor of three, and the flickering arises as a result of the approach to a critical value of external forcing of the climate system, in this case the gradually declining summer insolation during the Holocene. As the threshold is approached, small perturbations induced by the simulated variability in the model can cause the model to flicker between states[15].

In this work, we present a detailed statistical analysis of several wet–dry transitions in the 620,000-year environmental record from the Chew Bahir basin in the southern Ethiopian Rift. With its decadal resolution, the 1000-year-long termination of the AHP is particularly well documented, along with 20–80-year-long droughts, recurring every ~160 years, as possible early warnings. Together with events of extreme wetness at the end of the transition, these droughts form a pronounced climate "flickering", which can be simulated in climate models and is also present in earlier climate transitions in the Chew

Bahir environmental record, indicating that transitions with flickering are characteristic of this region.

## Results

### The Chew Bahir record of tipping points with precursors

The paleohumidity record from the Chew Bahir basin in the southern Ethiopian Rift documents the climate history of eastern Africa for the past ~620 kyrs[18] (Fig. 1). The climate in this part of Africa is determined by an interaction of several air streams and convergence zones, which —in combination with the varied topography, large lakes, and the nearby Indian Ocean—result in a complex spatiotemporal distribution of precipitation[19]. The dual passage of the tropical rain belt results in a bimodal distribution of rainfall in the Chew Bahir region today, whereby moisture reaching the Ethiopian highlands comes from the Mediterranean and Red Sea (55%), and from the Indian Ocean (31%)[20]. On annual to decadal time scales, the intensity of rainfall is related to the east–west adjustments in the Walker circulation associated with the Indian Ocean Dipole[19]. During the African Humid Period (15–5 kyr BP), the Chew Bahir basin was filled with a large freshwater lake up to the ~45 m overflow level to the Turkana basin[18].

In the years from 2009 to 2014, we recovered sediment cores, up to ~278 m long, from the southern part of the Chew Bahir basin[9,18,21,22] (Supplementary Table 1). High potassium (K) content in the sediment, determined by micro X-ray fluorescence (μXRF) scanning, was previously shown to be a reliable proxy for aridity in the Chew Bahir basin[9,21,23,24]. The major control on K concentrations is the hydrochemistry of the paleolake and porewater. These exert a direct control on the degree of authigenic mineral formation in the sediment, such as low-temperature illitization of smectites and analcime formation during episodes of higher alkalinity and salinity in the closed-basin lake resulting from a drier climate[23–26]. The progressive authigenic K-fixation in smectites with increasing evaporative conditions was found to be further enhanced by the excess in the octahedral layer charge that is caused by Al-to-Mg substitutions in clay minerals with more alkaline and saline lake conditions[23,24].

The decadal resolution of this proxy record provides an opportunity to examine the termination of the AHP and possible early warning signals[18,23]. Based on six well-dated short sediment cores (9–19 m, <47 kyr BP) and two long cores (292.87 m, <620 kyr BP) we studied the climate transition at ~5.5 kyr BP in detail, and similar transitions, including possible early warning signals before the first known occurrence of *Homo sapiens* on the African continent at ~318 kyr BP[27]. The Chew Bahir record is also particularly attractive because it is located close to the Ethiopian Plateau where rainfall feeds the sources of the Nile and other large rivers. The Chew Bahir record thus provides a high-resolution reconstruction of hydrological fluctuations that controlled living conditions along the Nile, and may provide insights into cultural innovation in this region[21].

At the end of the AHP in the short cores from Chew Bahir, we observe at least fourteen dry events, each 20–80 years long and recurring at 160 ± 40 years intervals (Fig. 2a). These dry events, interpreted as possible precursors of an imminent tipping point, would have allowed a prediction of climate change in the Chew Bahir basin at that time. Defining the actual tipping point in the record is difficult due to the presence of noise, but perhaps not crucial in assessing the transition and possible early warning signals (see "Methods"). Later in the transition, after ~6 kyr BP, seven wet events occur in addition to the dry events, with similar duration and recurrence rate. These high-frequency wet–dry extreme events represent a pronounced flickering in line with recent modeling results[15]. Interestingly, the older sediment record at Chew Bahir shows several transitions that are very similar to the termination of the AHP (Supplementary Fig. 3 and Supplementary Table 2). For example, the transition between ~382 and 376 kyr BP seems in its evolution extremely similar to the termination of the AHP and possibly points towards similar dynamics (Fig. 2b).

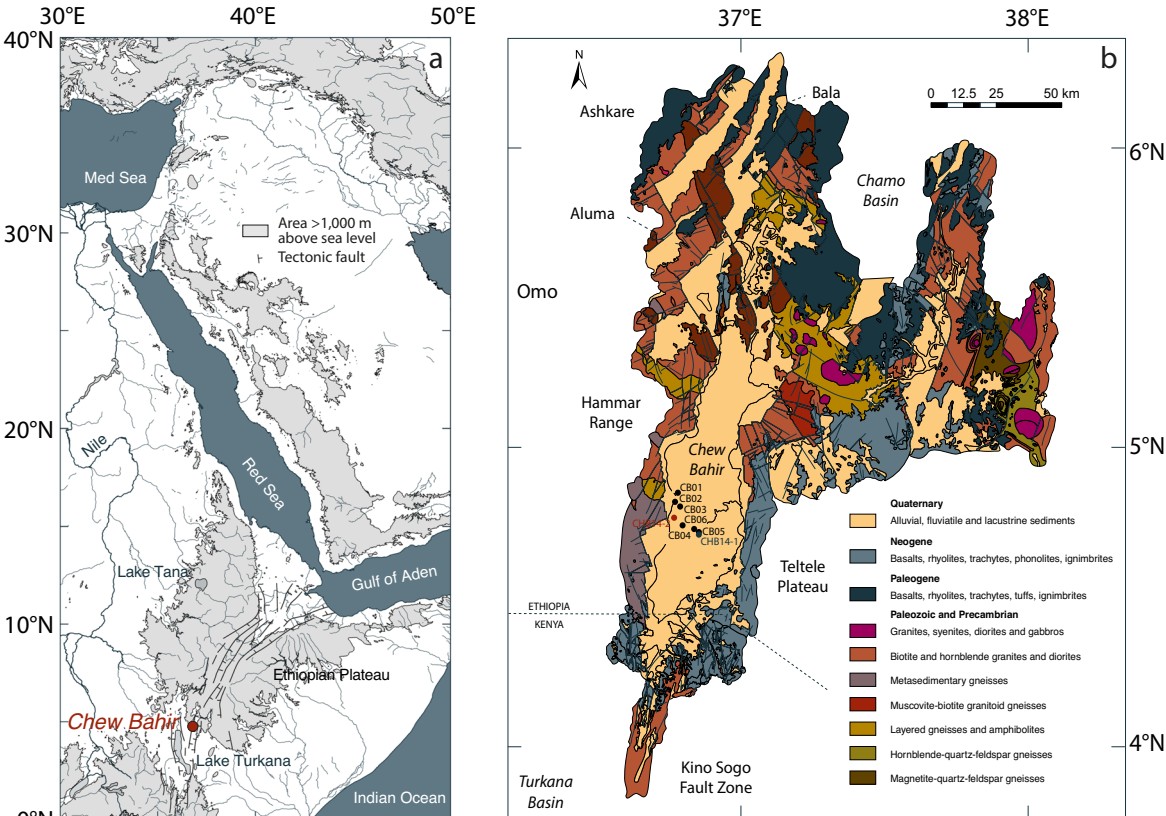

**Fig. 1 | Study area and localities of the lake cores. a** Map of northeastern Africa and adjacent areas showing the Ethiopian Plateau (in gray), the Ethiopian rift (marked with thin black lines), the Chew Bahir basin (4°45'40.55"N 36°46'0.85"E, ~500 m above sea level) and the river Nile with its two tributaries. Coastline and river polygons from the Global Self-consistent, Hierarchical, High-resolution Geography Database (GSHHG)[68]. Topography from the 1 arc-minute global relief model of the Earth's surface (ETOPO1)[69]. **b** Geological map of the Chew Bahir basin, showing the four generalized rock types: Quaternary rift sediments, Neogene and Paleogene rift volcanics, and Paleozoic–Proterozoic basement, and the location of the short cores CB01–06, the intermediate core CHB14-1 and the long cores CHB14-2A and 2B. Compilation based on Omo River Project Map[41], Geological map of the Sabarei Area[70], Geological map of the Yabello Area[71], and Geological map of the Agere Maryam Area[72].

## Validation of the tipping points in recurrence plots

The use of nonlinear methods, such as recurrence plots (RPs) and recurrence quantification analysis (RQA) reveals clear similarities in the dynamics contained in the time series (Fig. 2c–f and Supplementary Fig. 6). RPs are graphical displays of recurring states of a system, calculated from the (e.g., Euclidean) distance between all pairs of observations, (if required) within a cutoff limit (or threshold). A thresholded RP displays recurring states with a distance below the threshold value as black dots, while states above the critical distance are displayed as white dots. An unthresholded RP, not designated as such in some fields, is simply a pseudocolor plot of the distances between pairs of observations, therefore simply a graphical representation of the distance matrix. RQA uses measures of complexity for a quantitative evaluation of the RP's small-scale structures[28,29]. Among these, the recurrence rate (*RR*), indicated by the density of black dots in the RP, describes the propensity of the system to recur in a particular time period[28–30]. The ratio of the recurrence points that form diagonal structures (of a minimum length) is a measure for determinism (*DET*) of the system[28–30].

Exploring the RP/RQA results of the K record between 9 and 3 kyr BP reveals two major square blocky features in the RP, connected at ~5.5 kyr BP which coincides with the inflection point in the K curve (Fig. 2c). The two blocks differ significantly in their internal structure. The block between 5.5 and 3 kyr BP shows a very high density of recurrence points, which is reflected in high values of *RR* and *DET*. In contrast, the block between 9 and 5.5 kyr BP is, except for its older part between 9 and 8 kyr BP, characterized by both vertical and horizontal

lines, representing episodes of stability (both wet and dry) interrupted by a series of extremely dry events, indicated by white vertical lines in the RP. The interval between 8 and 5.5 kyr BP also shows remarkable diagonal features suggesting a cyclic recurrence of droughts in the Chew Bahir basin within a period that had a generally wet climate. The oscillating climatic conditions are reflected in higher *RR* and *DET* values indicating a relatively high predictability of climate, but much lower than before 8 kyr BP and after 5 kyr BP, both being episodes of relative stability and predictability, with extremely high *RR* and *DET* values close to one.

Examining the RP/RQA results of the modeled precipitation from a transient climate model simulation of the Holocene covering 10–0 kyr BP reveals very similar structures despite the differences due to the different character of the climate variable (i.e., sedimentary K concentrations versus simulated precipitation) (see "Methods" for more information about modeling) (Supplementary Fig. 6). There is also a blocky structure in the RP, but it looks different in detail because of the different course of the K curve and the modeled curve. The most striking similarity to the K curve from the Chew Bahir, apart from the rapid transition between 7 and 4 kyr BP, is the occurrence of diagonal structures in the range of the transition. Furthermore, it is striking that the spacing of the diagonals is also similar at about 150 years, which corresponds to the recurrence of droughts in the Chew Bahir. Thus, despite the differences in the characteristics of the climate variables, it looks like the model has captured a key feature of the tipping point at the termination of the AHP, including flickering.

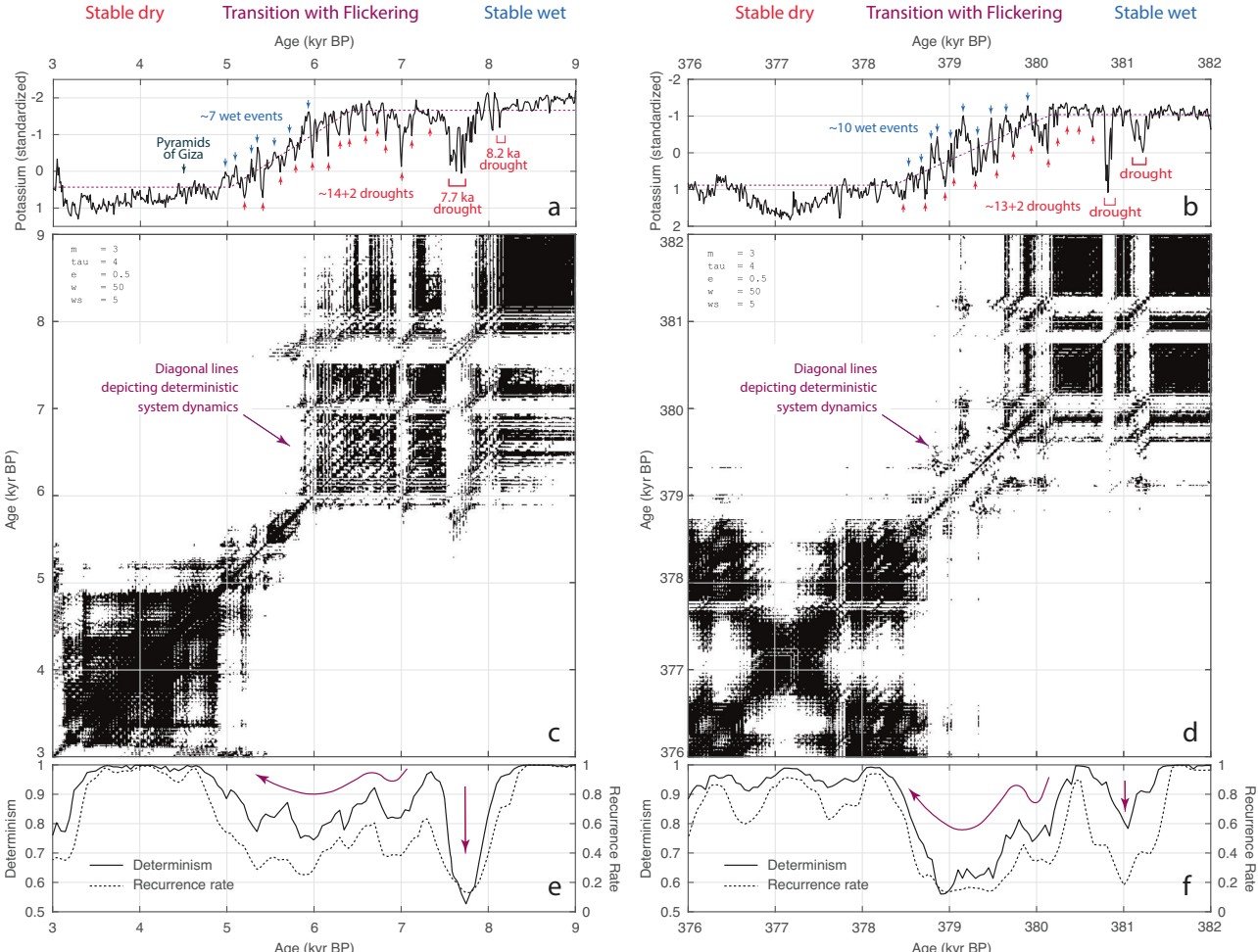

**Fig. 2 | Wet–dry transitions in the Chew Bahir during the past ~620 kyrs, recurrence plots, and recurrence quantification analysis results. a, b** Records of relative aridity in the Chew Bahir basin, southern Ethiopia, between (**a**) 9–3 kyr BP and (**b**) 382–376 kyr BP interval. During the past ~620 kyrs, climate in northeastern Africa passed multiple tipping points, for example, at ~7 kyr BP and ~380 kyr BP, respectively. After passing the tipping points, climate entered ~0.9–1.5 kyr long transitions from stable wet to stable dry climate, as described by nonlinear least-squares fitting a ramp function (dotted purple line) to the K curve from Chew Bahir. Both transitions are marked by pronounced flickering between the two extremes, wet (blue arrows) and dry (red arrows). **c, d** Recurrence plots (RPs) showing

remarkable diagonal features suggesting a pronounced flickering in the Chew Bahir basin after the tipping points, and **e, f** recurrence quantification analysis (RQA) measures *recurrence rate* (*RR*) and *determinism* (*DET*) of the Chew Bahir records with higher *DET* values indicating a relatively high predictability of climate, but much lower than before and after the transitions, both being episodes of relative stability and predictability, with higher *DET* values. See the supplementary information for more examples of wet–dry transitions with pronounced flickering in the ~620 kyr long climate record of the Chew Bahir, as well as for a detailed description and an interpretation of the RPs and RQA results.

Comparing the RP/RQA results of the K record between 9 and 3 kyr BP with those of older transitions, the one between ~382 and 376 kyr BP looks like an almost perfect replica of the termination of the AHP (Fig. 2). The RP of the transition at ~382–376 kyr BP shows the same blocky features, connected at ~379 kyr BP, and the diagonal lines, although not quite as clearly as in the RP of the K curve between 9 and 3 kyr BP, due to the cyclic recurrence of droughts in the range of the transition (Fig. 2c, d). Also, the curves of the *RR* and *DET* are very similar, with higher *RR* and *DET* values indicating a relatively high predictability of climate, but much lower than before 381.5 kyr BP and after 378 kyr BP, both being episodes of relative stability and predictability, with extreme *RR* and *DET* values (Fig. 2e, f). This is noteworthy considering that the older transition at 202.86 meters core depth (mcd) is documented in a core obtained with a different coring technique, and whose K concentration was measured with a different µXRF scanner. It also confirms K concentration as a very reliable climate proxy, even 379 kyr BP before today and, considering older similar transitions, before that (Fig. 2, Supplementary Fig. 3, and Supplementary Table 2). The similarity of the two transitions in

duration and structure further falsifies Wright's hypothesis[31] that humans, through pastoralism and agriculture, influenced the rate and structure of the climate transition in the mid-Holocene, at least for the region around the Chew Bahir. The transition at ~382–376 kyr BP is quasi-natural (without anthropogenic overprints) because larger populations of humans (namely *H. sapiens*) were absent from the region, in contrast to the transition at 9–3 kyr BP.

## Discussion
The flickering wet–dry events prior to major climate transitions, which seem to recur throughout the ~620 kyr Chew Bahir record, combined with the HadCM3 modeling results presented here, confirm the existence of precursor events prior to a tipping point, previously predicted only in theory. Although the Chew Bahir record offers what appears to be the clearest example of flickering, there are other examples, at least at the end of the AHP. For example, log(Ti/Ca) data from core MD04-2726 from the Nile delta deep sea fan, interpreted as a record of Nile flood events, show extreme droughts with similar duration and recurrence but about 1550 years earlier than at Chew Bahir[32]. Other

data sets showing excursions at the termination and flickering during the AHP include a lake record from Lake Dendi[33], a lake record from Lake Abiyata[34], and a record from Congo stalagmites[35].

The discovery of flickering prior to and during the transition from wet to dry conditions in Africa has significant implications for interpreting the relationship between climate change, cultural developments, and human migrations in those regions. After the hyperarid period coinciding with the high-latitude Younger Dryas (12.9–11.7 kyr BP) stadial, increased precipitation turned northeast Africa into a savannah-like environment with mean annual rainfall of about 50–150 mm[36]. Recolonization of this formerly desert area by small hunter-gatherer groups was probably a very slow process taking place over more than 1000 years when foraging ranges were gradually expanded[37]. During their seasonal rounds, ephemeral waterholes were one of the most important resources for these highly mobile groups, other than areas with permanent water access[38]. Living conditions remained harsh and must be assumed still to have been arid with high climatic variability, and patchy, unpredictable access to surface water. Therefore, people had to be able to adapt to changing and adverse living conditions, for instance, by flexible seasonal resource management, variable mobility patterns, and food sharing as risk minimization strategies[12].

By 7.3 kyr BP, the northeast African summer rain began to retreat southward in what was probably a gradual, rather than a stepwise shift[9,11]. During this period, hunter-gatherer groups began to leave areas far from permanent water availability. During the main flickering period with about fourteen short droughts, areas outside the Nile valley, the oases, and the few mountain refugia were already uninhabitable[12]. At ~7 kyr BP, many areas were abandoned due to climatic instability. We suggest that environmental instability caused by the climate flickering led to the abandonment of ancestral habitats, as conditions became too unpredictable to allow either nomadic or sedentary lifestyles. The discussion has too long centered on how fast these types of transitions have occurred (abrupt vs. gradual) and what stress this uni-directional change exerted on humans[9]. The flickering of climate, however, would have had a much more dramatic impact than the slow climate transitions spanning tens of generations. Confirmation of the existence of flickering events as precursors several times in the past, as shown by our data from Chew Bahir, may also provide insights into possible early warning signals for future large-scale climate tipping points.

## Methods

### Chew Bahir setting, materials and climate proxy

Chew Bahir (4.1–6.3°N, 36.5–38.1°E; WGS 84; Fig. 1) is a tectonic basin in the southern Ethiopian Rift with a ~32,400 km² catchment area. In the west, it is separated from the Omo Basin by the Hammar Range, which has become known through fossil findings of *Homo sapiens*[39,40]. In the western catchment, mainly Precambrian and Paleozoic granites, syenite, diorites, and gabbros occur, while in the eastern catchment, in addition to these rocks, Paleogene and Neogene basalts, trachytes and other volcanic rocks from the early phase of the formation of the rift dominate[41,42]. During the rainy seasons, the Chew Bahir mudflat is covered by water, mostly from the Weyto and Segen rivers entering the basin from the north. Forming a terminal sink for eroded material from the catchment since the Neogene formation of the Chew Bahir, the basin today contains more than 5 km of sediment[18].

In the years from 2009 to 2014, we recovered sediment cores of different lengths from the southern part of the Chew Bahir mudflat. In 2009 and 2010, we took six up to 19-m long cores CB01–06 along a NW–SE transect across the southwestern part of the basin[9,18,21,22] (Fig. 1 and Supplementary Table 1). Of these, CB01 and CB02 were recovered from the margin of an alluvial fan, an area that receives sediment primarily from the Hammar Range and the Weyto River. CB04–06, on the other hand, were collected further into the center of the Chew Bahir

and are supplied primarily by the Segen River from the north[9,21]. CB03 from an intermediate location along the transect, contains a fluvio-lacustrine mix of Weyto and Segen sediments[9,21]. In mid 2014, we collected the ~40 m CHB14-1 core in the central part of the basin, not far from CB05[22]. In late 2014, we collected the longest cores CHB14-2A and 2B, reaching down to 278.58 and 266.38 m depth, respectively[18].

The short cores CB01–06 and CHB14-1 were shipped to the U Cologne for further analysis and storage[9,18,21,22]. It is very important when evaluating micro X-ray fluorescence (μXRF) data and discussing possible measurement artifacts that the elemental content of CB01 were measured with a molybdenum (Mo) tube as radiation source in the μXRF scanning with an ITRAX core scanner, whereas cores CB02–06 were measured with a chromium (Cr) tube which has slightly different sensitivities towards elemental settings[9,18]. Due to the higher sedimentation rate at the basin margin, CB01 has a higher temporal resolution of a few years at the same spatial resolution of 0.5 cm, which is why short-term fluctuations such as the 20–80 year droughts are better represented in this core than in cores CB02–06 and CHB14-1[9]. The long cores CHB14-2A and 2B were shipped to the US Continental Scientific Drilling Facility (LacCore) for sampling, analysis and storage, except for μXRF measurements that were performed at the LacCore-associated Large Lakes Observatory (LLO)[18]. The shorter cores CB01–06 were dated by the radiocarbon method, recalibrated using IntCal20[43] for this work, and a composite age model was developed by linear and spline interpolation[9,18]. We used exclusively data from core CB01 for the analysis of the mid-Holocene tipping point to avoid artifacts in the analysis from stitching the composite of multiple short cores[9,29]. For this reason, the analysis is limited to the 9–3 kyr BP time interval, which does not show major fluctuations in the spacing of the data points, including gaps[9,29].

Core CHB14-1 was dated by radiocarbon (¹⁴C) dating and thermo-luminescence/optically stimulated luminescence (TL/OSL) age determinations; the age model was found using Bayesian age–depth modeling[22]. The long cores CHB14-2A and 2B were dated using ¹⁴C, OSL, and ⁴⁰Ar/³⁹Ar ages, and tephrochronological data, again using Bayesian age–depth modeling to develop the RRMarch2021 age model[9,44]. Cores CHB14-2A and 2B were spliced together to a common depth, forming composite core CHB14-2, using visual and physical sediment properties, resulting in 292.87 m long composite core with a core recovery of ~90%. The μXRF data sets were subjected to intensive quality control, which included outlier elimination, correction of offsets, and cleanup of duplicate values[9,18].

We compare the Chew Bahir data with simulated precipitation from a transient climate model simulation of Holocene covering 10–0 kyr BP[15]. This 10,000-year simulation employs the Hadley Centre coupled model version 3 (HadCM3)[45,46]. HadCM3 is a coupled three-dimensional atmosphere–ocean general circulation model with schemes for sea ice, and dynamic vegetation which is represented using a tiling of nine different land-surface covers (MOSES 2.1 with dynamic vegetation represented with TRIFFID, M2.1d)[47]. The horizontal resolution of the atmospheric model is 3.75° × 2.5° in longitude–latitude with 19 unequally spaced vertical levels. In the ocean the resolution is 1.25° × 1.25° with 20 unequally spaced vertical levels.

The University of Bristol configuration of this model (HadCM3B-M2.1d) is comparable to other more recent models in terms of skill in simulating present-day climatology[46]. In recent work three new configurations of this were developed, applying mid-Holocene-constrained parameter updates to the model's atmospheric convection (+CONV), vegetation (+VMS) or both (+CONV+VMS). The first three configurations[15]: standard (STD-equivalent to the Bristol configuration), +CONV, +VMS are unable to reproduce the mid-Holocene green Sahara convincingly, which is similar to the majority of other coupled climate models[48]. In this study we use the fourth configuration +CONV+VMS (or HadCM3BB-M2.1d)[15] because it shows a convincing

greening of the Sahara under mid-Holocene forcing and, when run transiently across the Holocene, it shows excellent agreement with precipitation reconstructions from northwestern Africa[49] against which it was not optimized. This version is the only configuration to display flickering and has recently convincingly reproduced the timing of multiple green Sahara phases back to 800 ka BP[50].

For the Holocene, the HadCM3BB-M2.1d was forced with changes in Earth's orbit[51], the time-varying distribution of ice sheets, land area and sea level from ICE-6G[52,53] and mixing ratios of carbon dioxide, methane and nitrous oxide as reconstructed from ice cores[54–56]. Variations in tropospheric aerosols are not included, but volcanic eruptions, the total solar irradiance and land use have been analyzed separately and do not markedly alter the Holocene simulation over northern Africa[15]. Volcanic stratospheric aerosol optical depth is prescribed in the model based on the results from HolVol v1 bipolar reconstruction[57] and total solar irradiance has been reconstructed using cosmogenic isotopes[58]. In this study, we analyze the climate simulation forced with Holocene variations in orbit, ice sheets, greenhouse gases, solar irradiance and volcanic eruptions and using the paleo-conditioned configuration, HadCM3BB-M2.1d[15,59].

## Validation of the tipping points in statistical parameters

The K curve of Chew Bahir core CB01 was analyzed section by section (within ~6-kyr windows) using methods from nonlinear dynamics such as recurrence plots (RPs) and recurrence quantification analysis (RQA) (Fig. 2 and Supplementary Figs. 4–6). These methods require evenly-spaced time series, which is why we first converted the data from composite depth to age using a previously published linear age model[60]. Subsequently, the data were interpolated to an evenly-spaced time axis running from 45.358 to 0 kyr BP at 10-year intervals using a shape-preserving piecewise cubic interpolation[61] implemented in the MATLAB function `pchip`.

Recurrence plots (RPs) are graphical displays of recurring states of a system, calculated from the (e.g., Euclidean) distance between all pairs of observations, (if required) within a cutoff limit[28–30,62]. The visual inspection of RPs is often complemented by a recurrence quantification analysis (RQA), which uses measures of complexity for a quantitative evaluation of the RP's small-scale structures[28–30]. Among these, the *recurrence rate* (*RR*) is measuring the density of black dots in the RP, describing the propensity of the system to recur in a particular time period[28–30]. Diagonal lines in RPs are diagnostic of predictable behavior in time series and can be used to predict future conditions from the present and past. The ratio of the recurrence points that form diagonal structures (of a minimum length) to all recurrence points is a measure of *determinism* (*DET*) of the system[28–30].

For the analysis of the 9–3 kyr BP record, the K record from the short cores was embedded in a phase space with a dimension of $m = 3$ and temporal distances of $\tau = 4$ (Supplementary Fig. 4), equivalent to $4 \times 10$ years = 40 years, where 10 years is the resolution of the time series following a piecewise cubic Hermite polynomial interpolation[63]. We use the window size $w = 50$ and the step size $ws = 5$ data points of the moving window to calculate the RQA measures. The size $w$ of the window corresponds to $50 \times 10$ years = 500 years and the step size is $5 \times 10$ years = 50 years. We use a minimum length of 10 points to compute *DET*. To compare the RP/RQA-based dynamics in the Chew Bahir record of aridity and the modeled precipitation record, we interpolated the modeled record to the same time axis, used the same embedding parameters to create the RPs and used the same window size to calculate the RQA measures (Supplementary Fig. 6). Similarities in the texture of the recurrence plots of both proxy records show that the embedding provides comparable results with these values.

## Determining the duration and start of tipping

We statistically analyzed the K curve from the Chew Bahir for the amplitude, duration, and midpoint of a climate transitions between 9

and 3 kyr BP and between 382 and 376 kyr BP using three different methods: sigmoid fit, ramp fit and change point detection[9,64] (Supplementary Fig. 7). First, we applied a change point search algorithm[61] to the standardized K record, i.e., the mean K values were subtracted from the individual K values and then divided by the standard deviation. The algorithm, which has been included in MATLAB as the `findchangepts` function, detects change points by minimizing a cost function over all possible numbers and locations of change points. The `findchangepts` function yields the number of significant changes in the mean, the standard deviation, and the trend of a time series (not exceeding a maximum number of permissible changes defined by the user) that minimize the sum of the residual error and an internal fixed penalty for each change. Second, we fitted a sigmoid function with four parameters *a*, *b*, *c*, and *d*

$$x_{\text{fit}}(t) = a + \frac{b}{1 + e^{-d(t-c)}}, \text{for} -\infty < t < +\infty \quad (1)$$

to the records $x_i$ with $1 < i < n$ and *n* data points using nonlinear least-squares fitting. The sigmoid function (in its normalized representation) is a monotonic s-shaped curve, often referred to as a smooth version of a step function. The sigmoid function is bounded by two horizontal asymptotes $x_{\text{fit}}(t) \to 0$ and 1 as $t \to -\infty$ and $+\infty$, respectively. It has a bell-shaped first derivative curve and exactly one inflection point (parameter *c*), which can be interpreted as the midpoint of the transition. In our analysis, we used the function `fit` together with `fitoptions` and `fittype` included in the Curve Fitting Toolbox of MATLAB to fit a sigmoid function to the data (Supplementary Fig. 7). Third, we statistically re-analyzed these records, fitting a ramp function again with four parameters $x_1$, $x_2$, $t_1$, and $t_2$

$$x_{\text{fit}}(t) = \begin{cases} x_1, & \text{for} \quad t \le t_1 \\ x_1 + \frac{(t-t_1)(x_2-x_1)}{t_2-t_1}, & \text{for} \quad t_1 < t \le t_2 \\ x_2, & \text{for} \quad t > t_2 \end{cases} \quad (2)$$

to the *n* data points. The monotonic ramp-shaped curve has two horizontal pieces and an inclined piece, connected by two abrupt changes of direction and with a discontinuous first derivative. Both the sigmoid and ramp functions are widely used to describe transitions in climatic and environmental conditions as well as the response of the biosphere[61,65]. Again, we used the function `fit` together with `fitoptions` and `fittype` included in the Curve Fitting Toolbox of MATLAB to fit a ramp function to the data (Supplementary Fig. 7). Using the methods do describe the transitions, we found the following measures for the transition between 9 and 3 kyr BP

```
changepoints =
  5.6950
sigmoidfit =
  General model:
  sigmoidfit(x) = a + b*(1./(1+exp(-d*(x-c))))
  Coefficients (with 95% confidence bounds):
    a =  0.4223 (0.3727, 0.4719)
    b = -2.053 (-2.127, -1.979)
    c =  5.722 (5.66, 5.784)
    d =  4.178 (3.238, 5.118)
rampfit =
  General model:
  rampfit(t) = rampfunction(t,t1,t2,x1,x2)
  Coefficients (with 95% confidence bounds):
    t1 =  5.064 (4.951, 5.177)
    t2 =  6.37 (6.256, 6.483)
    x1 =  0.4175 (0.3691, 0.4659)
    x2 = -1.623 (-1.674, -1.572)
```

resulting in the lower and upper knick points of the ramp at $t_1 = 5.1$ kyr BP (corresponding to ~3.71 mcd) and $t_2 = 6.4$ kyr BP (corresponding to ~4.30 mcd) (using a previously published linear age model[60], but with radiocarbon ages recalibrated to IntCal20[43]). For the transition between 382 and 376 kyr BP, we found

```
changepoints =
  379.0600
sigmoidfit =
  General model:
  sigmoidfit(x) = a + b*(1./(1+exp(-d*(x-c))))
  Coefficients (with 95% confidence bounds):
    a = 0.8967 (0.8415, 0.952)
    b = -1.9 (-1.985, -1.816)
    c = 379.2 (379.1, 379.3)
    d = 3.186 (2.481, 3.89)
rampfit =
  General model:
  rampfit(t) = rampfunction(t,t1,t2,x1,x2)
  Coefficients (with 95% confidence bounds):
    t1 = 378.4 (378.2, 378.5)
    t2 = 380.2 (380, 380.3)
    x1 = 0.8868 (0.8346, 0.939)
    x2 = -1.013 (-1.07, -0.956)
```

resulting in the lower and upper knick points of the ramp at ~380.2 kyr BP (corresponding to ~203.5 mcd in the core) and 378.4 kyr BP (corresponding to ~197.7 mcd) (using the age model RRMarch2021[44]). It is tempting to mark the upper knick point $t_2$ of the ramp as the most likely time for the climate tipping. However, the fitting of a ramp but also other methods for determining the tipping point[66,67] use only static approaches to define a knick point, but can never determine the actual tipping. Furthermore, the flickering in our type of tipping point, as well as the presence of noise, makes it difficult to precisely define a tipping point in a proxy record. The visual inspector might be tempted to mark the tipping point in the younger transition at ~5.9 kyr BP, and perhaps at 379.2 kyr BP in the older transition, i.e., at the last wet events in each case with an amplitude that reaches the level of K values in the stable wet phase before the transition. But perhaps the precise definition of the beginning of tipping is not crucial. It is much more interesting to see that the climate starts to flicker well before the actual transition (e.g., more than one thousand years earlier in our examples) and hence the tipping elements sent out clear early warning signals in the form of 20–80-year-long extreme events with clear regularity in recurrence.

## Data availability
The data that support the findings of this study are available from Zenodo with the identifier https://doi.org/10.5281/zenodo.10624471 (https://zenodo.org/records/10624471).

## Code availability
The MATLAB code to recreate the figures are available from Zenodo with the identifier https://doi.org/10.5281/zenodo.10624471 (https://zenodo.org/records/10624471).

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

## Acknowledgements

Support for the Hominin Sites and Paleolakes Drilling Project (HSPDP) led by Andrew S. Cohen has been provided by the National Science Foundation (NSF) grants and the International Continental Drilling Program (ICDP). Support for Chew Bahir Drilling Project (CBDP) led by A.A., H.L., F.S. and M.H.T has been provided by Germany Research Foundation (DFG) through the Priority Program SPP 1006 ICDP (SCHA 472/13 and /18, TR 419/8, /10 and /16), the CRC 806 Research Project "Our way to Europe" Project Number 57444011 and the UK Natural Environment Research Council (NERC, NE/K014560/1, IP/1623/0516). We also thank the Ethiopian permitting authorities to issue permits for drilling in the Chew Bahir basin. We also thank the Hammar people for the local assistance during drilling operations. We thank DOSECC Exploration Services for drilling supervision and Ethio Der Pvt. Ltd. Co. for providing logistical support during drilling. Initial core processing and sampling were conducted at the US National Lacustrine Core Facility (LacCore) at the University of Minnesota. We thank Christopher Bronk Ramsey, Melissa S. Chapot, Alan L. Deino, Christine S. Lane, Helen M. Roberts, and Céline M. Vidal for discussions on geochronology and age modeling. S.K.B. has received further financial support from the University of Potsdam Open Topic Postdoc Program. This is publication #55 of the Hominin Sites and Paleolakes Drilling Project.

## Author contributions

M.H.T., A.A., H.F.L. and F.S. designed the Chew Bahir Drilling Project. A.A., H.F.L., F.S. and V.F. led the drilling campaign. V.F., H.F.L., A.A. and F.S. measured and sampled the cores. M.H.T. and N.M. designed and ran the time series analysis experiments. P.O.H. and P.J.V. performed modeling experiments. M.H.T. led the writing of the paper and wrote the first draft of the text, including the paragraphs on time series analysis and the determination of the duration and start of the tipping. V.F. and M.H.T wrote the paragraph on validating climate tipping in the sediment in the methods section. M.H.T., M.L.F. and N.M. wrote the paragraph on recurrence plots and recurrence quantification analysis in the methods section. K.K. and M.H.T. wrote the paragraphs on the implications of climate tipping for humans in the main text. M.H.T., A.A., H.L., F.S., S.K.B., M.L.F., M.A.M. and V.F. edited the Introduction of the main text. M.H.T. designed all the figures in the paper. M.H.T., A.A., M.L.F., P.O.H., V.F., S.K.B., K.K., H.F.L., N.M., M.A.M., F.S. and P.J.V. discussed the results and contributed input to the manuscript.

## Funding

## Competing interests

The authors declare no competing interests.
