## [Peer Review File · Nature Communications]

Early warning signals of the termination of the African Humid Period(s)REVIEWER COMMENTS

Reviewer #1 (Remarks to the Author):

The manuscript by Trauth and co-authors targets the structure and patterns of the transition out of the African Humid Period (AHP) towards drier conditions in northern Africa using element geochemical records (K intensities), which sensitively record hydrological changes in lake sediments with robust chronological control from Chew Bahir, Ethiopia. The main focus of the study is on understanding early warning signals of climate tipping points in the African monsoon system. Remarkably, the study reveals a 'flickering' marked by abrupt alternations between dry and wet spells in both recent and older Pleistocene transitions out of AHP's. This flickering, distinct from preceding and subsequent periods, aligns with predictions from climate modeling studies, suggesting it as a precursor to climate tipping points in the African monsoon system. The manuscript employs quantitative data analysis, including recurrence plots and recurrence quantification analysis, on both elemental and modeling time-series, yielding comparable results. Consequently, the assertion that flickering is an intrinsic feature of an AHP tipping point is grounded in thorough data analysis and reasoning. The study also contextualizes hydrological changes by examining their implications on human environments. It argues that the flickering, characterized by brief yet intense dry spells, could have been a significant stressor influencing sedentary and nomadic human migrations during the examined periods. As such, this study is also an important outcome of the HSPDP scientific drilling program.

Overall, I recommend some moderate revisions on the manuscript prior publication and would particularly like to suggest the authors expand more thoroughly on the mechanism behind the flickering during the transitions. Especially, the following questions need to be addressed in the manuscript:

- It remains unclear whether the flickering is site specific to Chew Bahir with its near equatorial position and rainfall being controlled by the twice annual passage of the ITCZ. The authors refer to nearby Lake Dendi but at this location it seems as if the 'flickering' occurs more prominently during the AHP rather than within the transition out of the AHP. The records from Lake Abiyata and the Congo Basin are too low in resolution to document a flickering similar to that observed at Chew Bahir. Is the flickering also apparent in climate model time-series for the more classical North African Monsoon region to the N and W? What area (longitude, latitude) is actually included in the simulated precipitation time series shown in Suppl. Fig. 6? Elaborating on this is important, especially in light of the discussion of human environments and migrations.
- What, from a mechanistical perspective, exerts the major influence of the development of the wet and dry spells? Is this related to a change in the position of the Walker Cell in response to more stationary/persisting negative/positive IOD phases? Strengthened/weakened intrusion of the N African Monsoon into E Africa? Shifts in the Congo Air Boundary? Providing conceptual models and the synoptic climate scenarios for the flickering is important and will contribute to a better understanding of climate dynamics in this complex setting.

Using the HadCM3 output it should be possible to expand somewhat on the questions raised above. In my opinion, while acknowledging the limitations inherent to computational models, this would further increase the significance of the statements and conclusions made.

Technical comment:

- Reference 30 is not an appropriate reference for Lake Dendi.

Reviewer #2 (Remarks to the Author):

The paper focuses on short term fluctuations in the concentration of K⁺ in the Chew Bahir record that occurred between 9-3 ka and during other terminations throughout the 620ka record. The paper suggests that these fluctuations are episodes of climatic flickering that precede the African Humid Period termination and if these are a general characteristic of tipping points in Earth's climate they could serve as early warnings of future tipping points.

Overall, the paper is very interesting and the results could have important implications regarding the dynamics of tipping points in Earth's History and particularly to the termination of the African Humid Period. I commend the authors for the work they have done and the intriguing results they present. I think the results and more importantly the discussion of flickering in the context of tipping points in Earth's history should be published in NC.

However, I have a few major issues with this paper. The first (points 3a and 3b) regard the question of whether these short K⁺ fluctuations are real climatic signals or rather intrinsic noise or result from sedimentary dynamics. The climatic origin of these fluctuations needs to be laid out in a more convincing way. The second issues (points 1 and 2) regards the overall structure and main claims and discussion of the paper.

I think this paper can become a NC paper, but it must provide a more compelling argument regarding the climatic origin of the K signal and a more detailed mechanistic understanding of these flickering events in the overall context of the termination.

Major

1. I am not sure why you define the AHP termination as a tipping point. The current record shows a gradual ~1500 yr drying (from 6.5 to 5 ka). This is faster than the insolation decline, but arguably is not as fast as D-O and H events and the onset of the Holocene. In previous papers from this group (e.g., Truth et al., 2018) the rapidness of the termination is more robust. Given a 20 year debate regarding the gradual vs. abruptness of the AHP termination, these definitions are of importance. It would be useful to explain what exactly you define as a tipping point, and why there are differences between the different papers regarding the rapidness of the decline. In the supplementary data you go into great length to determine the time of the tipping point, but there is not mention of this in the main text. It would be beneficial to acknowledge the analysis in the main text.

This is not a "real" major point, but as the definition of this event as a tipping point lies at the heart of this paper, it is crucial to clarify this point in more detail.

2. What is the mechanistic relation between the fluctuations and the termination? The fluctuations occur over a time period of 2.5 ka and it is unclear why they occur before, during and after the decline of the AHP. In some places you say the fluctuation are precursors to the termination, but they occur throughout. Currently, the paper reads as an interesting observation, but does not provide a mechanism or links the short-term fluctuations with a better understanding of the termination. This point needs to be clearer.

3a. The main hypothesis of this paper is that the small fluctuations are real climate signals. Why are they not noise? Even the pronounced D-O events in GRIP were not believed until the publication of GISP2 that showed the same patterns (Grootes et al., 1993, Nature). I suggest removing the long-term average and looking at a time series of the variance in different size averaging windows. Is the variance between 7-5ka statistically larger than the average noise throughout the Chew Bahir record? Then it would be worth checking whether the size of the variance is a function of sedimentation rate, i.e. does higher degrees of noise occur in higher sed. rate portions of the core? I think this is clear when comparing the large variance between 0-10 ka and 35-45ka when sed. rate is high with 35-10ka when sed. rate is low and variance is low as well (from Foerster et al., 2014).

I'm not sure the method I suggested is the way to go, but for this paper to be published in NC, I think you must find a way to convince that you are not over interpreting noise.

3b. The fluctuations that are at the center of this paper are interpreted as representing a climatic signal. However, I think that internal sediment dynamics could be an additional possible source of these fluctuations. I'm going to play the devil's advocate in the hopes that this will help make the

climatic interpretation more robust.

An alternative source of these fluctuations is that as the lake is drying out, older lacustrine sediments become exposed at the flanks of the lake and get washed into the lake and accumulate at the margin. Mixing of early Holocene sediments (high K+) with the new ~5.5ka formed sediment (low K+) is a potential, non-climatic source of this flickering.

The fluctuations appear most prominently in the core from the margin of the lake and not in those from the center of the lake. The authors explain this observation by an enhanced sedimentation rate at the margin. However, from Foerster et al., 2014 it seems that CB01, CB02 and CB06 have the same sedimentation rate. So, if the fluctuations are climatically driven they should have shown up in both the margin core and the center of the lake core, but they show up only in the margin core, which might strengthen a sedimentary mixing origin of these fluctuations.

In addition, if the fluctuations formed due to sedimentary mixing, then the 14C ages throughout this time interval should also be mixed. The sed. rate for CB01 in Foerster et al., 2014 shows that the sediment rate is not linear during this time interval and that there are potentially large changes in the sedimentation rate (a rapid sedimentary rate increase around 7ka and then a drop in sedimentation rate after that). These are smoothed out when using the linear model compilation. This rapid change in sedimentary rate could also be an important clue strengthening the sedimentary mixing origin of the signal.

This alternative mechanism should be explored, and the two observations need to be addressed before accepting the interpretation of the flickering as having a climatic origin.

Line comments

L31. The abstract boils down to a local phenomenon and it is unclear why this flickering is of importance to a wider audience. Are there mechanistic reasons for this flickering? Do you expect this to occur in other regions? etc.

L71-79. This seems like a very interesting point, but throughout the whole paragraph is unclear what are the possible types of tipping point for the AHP. The two options in lines 69-70 are unclear in respect to the AHP termination. I suggest elaborating and maybe presenting hypothesis for what you would expect to see for different types of tipping points. You mention the tipping point models in passing a few times in the paper, so I think it's worth to say a little bit more about what model these are and what conclusions you draw from them.

L76-77. The model results of Hopcroft and Valdes 2021 show a large drying events at 7.7 ka which they claim could be a precursor to the collapse. This is very different from a series of small fluctuations you see in your record. So, I'm not sure "predicted by one recent modeling study" is an accurate statement.

L91. I suggest comma after Turkana basin and full stop after catchment.

L108-109. This line is basically the bottom line of the whole paper and I'm having a lot of trouble with it. The low K+ events occur before, during and after the termination, so I'm not sure why you claim that they are a precursor. Maybe they indicate that: 1) the termination wasn't so abrupt and 2) that the transition is characterized by enhanced variability. In addition, it seems like you are using the term "tipping point" as a specific point ("imminent") in time, but the record shows a 1500 year decline.

L111-112. What do these models show? It would be useful to add one sentence to explain, otherwise this is a pretty obscure sentence.

L115. What MIS is this?

L123. In the methods section (L237) and caption figure line 618 the duration of CB01 is 9-4 ka, which is inconsistent with this sentence.

L125. Do you interpret this as the tipping point occurring at 5.5ka? you should say this explicitly and refer to the supp material.

L160. I assume you mean the termination of the Holocene AHP?

L166. What modeling results are presented here?

L192. I think there is a contradiction here. You say (L189) that at 7 ka the site were abandoned, but here you say that the flickering starting at 6.5 ka is what caused the abandonment. Please explain.

L227. From the Foerster et al., 2014 it seems that CB01, CB02 and CB06 have the same sedimentation rate. This data seems contradictory to this statement. Please explain.

L247. Where do you compare the record to the model? I didn't find this in the paper. Do you mean on line 111-112? If so, this is very unclear.

L247-255. The Hopcroft and Valdes 2021 model was run using three different model configurations (standard, enhanced convection and dynamic vegetation). I assume you used the standard? Could you say a word on why you chose this run as opposed to the other two options?

L284. See comment on line 123.

Reviewer #3 (Remarks to the Author):

A really interesting and well-written manuscript. The end of the African Humid Period is a classic example of a climate tipping point which is well-known within the palaeo community. The manuscript explores the potential for identifying 'early warning signals' of this abrupt change.

I struggled initially to work out what was new in this paper compared to previous papers published by the authors. The Itrax data and chronology have all been previously published for the short cores (CB01-06) but the data for the long cores is new. The recurrence quantification analysis has also been published for the short cores, which possibly explains the lack of detail in explaining the method. I found I had to read Trauth et al. (2019), which provides some excellent figures of how the recurrence quantification analysis works. I do think that some more explanation of the recurrence quantification analysis is needed in this paper – or at the very least it is essential to reference the Trauth et al. (2019) paper earlier in the manuscript. At the moment it is only referenced in the Methods; it needs to be added somewhere in the section from line 117. Altogether, there needs to be a bit more transparency in what has been previously published, and what is new here. There is certainly enough to warrant a separate paper, but details from previous work either needs to be explained again, or need to be clearly signposted to find that information.

The manuscript mentions that the Chew Bahir record shows many of these transitions from dry to wet. It is not clear how many of these show this 'flickering' behaviour. Only two time periods appear to have been analysed – the Holocene and 382-376 kyr BP. Why only these two? It would be interesting to add some (brief) discussion regarding how many of the transitions back in time show the flickering behaviour, and for those that don't, why don't they? Is it a data resolution/proxy problem, or is flickering not apparent? I am also not sure how the wet-dry transitions highlighted in Suppl. Fig. 3 were chosen (A-K). There are several other periods of time that show an abrupt change in potassium, but are not highlighted. The figure caption states that the 10 examples 'differ in the occurrence of flickering and early warning signals', but I would like some more detail on this.

Given the journals broad audience I would consider whether all the acronyms are needed, and whether some can be spelt out. The section from Line 117 is very heavy on the acronyms, which makes it difficult to follow. I don't think DET is ever spelt out – presumably this is Determinism.

I am surprised that there are only 2 figures, but they are highly detailed. Figure 1 from Trauth et al. 2019 contains a much better map for panel A – in this figure it is difficult to tell where the site is located in relation to the plateau. Figure 2 would probably benefit from having the y-axis labelled to help to decipher the figure for anyone not familiar with recurrence quantification analysis.

Overall the results are intriguing and the likeness of the two transitions (Holocene and 382-376 kyr BP) is quite remarkable and certainly worthy of publication.

Technical comments

Line 95: Sentence fragment.

Line 107: I would reverse the sentence to first say 'nearing[?] the end of the AHP in the short cores from Chew Bahir, we observe.....' Makes it clearer on first read what time period is being discussed.

Line 108: strangely written... 'would have allowed a prediction of climate change'. By who? Or do you mean that we can use them to postdict? I'm not sure predication is necessarily the most helpful word here.

Line 110: This sentence talks about 'later' but it is not clear later from what time. I think there needs to be first some discussion of the nature of the transition – the idea of a tipping point suggests that it is step change, but the record shows that this transition did occur over a period of time (but faster than the forcing, which is the key point). The ramp fit analysis is good but it is never mentioned in the main manuscript.

Line 119: in the time series over what period of time? The whole record?

Line 161: what do you mean by quasi-natural?

Line 165: this sentence seems to imply this paper is the first to show real world data that supports the hypothesis that there can be identifiable early warning signals of climate tipping points. Of course, there are many papers that have shown real world data to support this notion (some even on palaeoclimate data from other monsoon systems e.g. East Asian and West African monsoon). Please rephrase this sentence, or add appropriate references.

Line 190: You say at ~6.5 ka but then describe this as a period. If talking about a period of time please include a start and end date.

Line 224-226: Why were some cores measured with Mo tube and others a Cr tube? Was it a deliberate choice or simply due to the location of the cores?

Line 222 down: It is quite confusing to work out what CB01-06 refers to. Perhaps you can add the words 'composite core' as otherwise it is necessary to look through the supplementary material to understand the core naming. Perhaps the word 'sequence' can be used for multiple cores in the same hole, and the word 'core' for a single drive.

Line 233: Please make clear that the age model is from ref. 63 but recalibrated using IntCal20 (as clarified in line 270),. It is a bit surprising that a linear age model is used when there are much more sophisticated Bayesian models widely available, but I acknowledge that this would be unlikely to affect the results.

Line 234 and Line 242: Strange use of the word 'find'. 'Develop' an age model?

Line 244: what datasets? The Itrax data?

Line 267: rephrase 'was subjected to'

Line 270: how did you choose the 10yr interpolation window? What is the average timestep of the data? Was there missing data? Were there some periods where this may have overinterpolated the data?

Paragraph from line 274 (and corresponding line in main text): this is not sufficient detail. What is meant by pairs of measurements? What is the cut-off limit? There needs to be a clearer explanation of

how this works. Similarly, the recurrence rate explanation is not clear. Over what time period does it measure the density of black dots? For each time step of 10 years?

Paragraph from line 296: Description of the change point/ramp fit analysis is well explained. But how it is used in the analysis is not clear. It is true that defining a particular point of tipping is not necessarily meaningful. But does the flickering occur before the tipping or is it part of the tipping? I.e., once a system starts to flicker, is the tipping locked in, or is it reversible? This seems to be the most interesting question.

Reviewer #1

Reviewer: Overall, I (...) would particularly like to suggest the authors expand more thoroughly on the mechanism behind the flickering during the transitions. (...) It remains unclear whether the flickering is site specific to Chew Bahir with its near equatorial position and rainfall being controlled by the twice annual passage of the ITCZ. The authors refer to nearby Lake Dendi but at this location it seems as if the 'flickering' occurs more prominently during the AHP rather than within the transition out of the AHP. The records from Lake Abiyata and the Congo Basin are too low in resolution to document a flickering similar to that observed at Chew Bahir.

Authors: Of course, we agree with Reviewer #1 that flickering, as observed at multiple transitions in the Chew Bahir, is not as clear in the other data series cited, for reasons given by the reviewer. However, it is common practice and, in our opinion, perfectly permissible for a phenomenon newly observed both in modeling results and for the first time in empirical data to be searched for in other data. We hope that this will inspire colleagues working on these other data series, and possibly a few more, to take a second look at their data. Perhaps they will even return to their data or samples (for example in Lake Abiyata or the Congo Basin) to re-examine the climate archives at a higher resolution. In fact, we are excited about to contribute with our research to a classic scientific novelty: first, a hypothesis is formulated and published. Then, it is tested until completely or partially falsified and replaced by a new hypothesis. In our study, we have already completed this step, by searching for a phenomenon we had already identified for the mid-Holocene. Identifying several examples of very same phenomenon, much further back in time, in a different core, obtained with a different drilling technique, measured with a different XRF scanner, – and yet showing a very resemblance to the flickering pattern in the mid-Holocene transition, seems very convincing to us.

Reviewer: Is the flickering also apparent in climate model time-series for the more classical North African Monsoon region to the N and W? What area (longitude, latitude) is actually included in the simulated precipitation time series shown in Suppl. Fig. 6? Elaborating on this is important, especially in light of the discussion of human environments and migrations. What, from a mechanistical perspective, exerts the major influence of the development of the wet and dry spells? Is this related to a change in the position of the Walker Cell in response to more stationary/persisting negative/positive IOD phases? Strengthened/weakened intrusion of the N African Monsoon into E Africa? Shifts in the Congo Air Boundary? Providing conceptual models and the synoptic climate scenarios for the flickering is important and will contribute to a better understanding of climate dynamics in this complex setting. Using the HadCM3 output it should be possible to expand somewhat on the questions raised above. In my opinion, while acknowledging the limitations inherent to computational models, this would further increase the significance of the statements and conclusions made.

Authors: The simulated timeseries shown in Figure S6 is the northern hemisphere summer (JJA) precipitation averaged over the area of maximum variability in the model which is the North Western part of Africa (20–30°N by 20°W–5°E). We have added this information to the figure caption.

This region corresponds to the regions of flickering analysed in previous studies (Hopcroft and Valdes, 2021, 2022). In these studies, we found that the model displays strong positive feedbacks between vegetation cover and precipitation, in agreement with theoretical predictions (e.g. Charney et al., 1975). This is caused by both radiative and hydrological effects of vegetation. Firstly, the darker vegetation enhances solar energy absorbed fueling monsoonal type circulation onto land and hence precipitation. Vegetation as opposed to bare soil enables more efficient moisture-recycling back to the atmosphere, thus providing a pair of self-reinforcing feedbacks. In this configuration of the climate model, we find that these two effects are approximately equal in strength. The flickering arises as a result of the approach to a critical value of external forcing of the climate system. In this case the gradually declining summer insolation during the Holocene. As the threshold is approached, small perturbations induced by the simulated variability in the model can cause the model to flicker between states (Hopcroft and Valdes,

2021). We found this behavior to be independent of the initialisation of the model (e.g. the state of the ocean) and instead, it is primarily determined by these two positive feedbacks (Hopcroft and Valdes, 2022). This information is all available in previous papers and as noted above, we refer readers of the current work to those papers rather than including lots of this material in this work.

Also please see out response to Review #2 on their comment on L71-79.

Reviewer: Reference 30 is not an appropriate reference for Lake Dendi.

Authors: Done. The correct reference is Wagner et al. (2018).

Reviewer #2

Reviewer: (...) I am not sure why you define the AHP termination as a tipping point. The current record shows a gradual ~1500 yr drying (from 6.5 to 5 ka). This is faster than the insolation decline, but arguably is not as fast as D-O and H events and the onset of the Holocene. In previous papers from this group (e.g., Trauth et al., 2018) the rapidness of the termination is more robust. Given a 20 year debate regarding the gradual vs. abruptness of the AHP termination, these definitions are of importance. It would be useful to explain what exactly you define as a tipping point, and why there are differences between the different papers regarding the rapidness of the decline. In the supplementary data you go into great length to determine the time of the tipping point, but there is not mention of this in the main text. It would be beneficial to acknowledge the analysis in the main text. This is not a “real” major point, but as the definition of this event as a tipping point lies at the heart of this paper, it is crucial to clarify this point in more detail.

Authors: As the termination of the AHP meets all the criteria for representing a classic example of a tipping point (as in detail set out in the first section of the paper including the corresponding literature cited there), we were somewhat surprised by the reviewer’s hesitation to classify the AHP termination as a tipping point. In fact, we also read in the comment of Reviewer #3, "The end of the African Humid Period is a classic example of a climate tipping point which is well-known within the paleo-community." As we say in L194–195 of the original version of the paper, there has been a long debate about whether the end of the AHP is abrupt or gradual, we agree and published specifically on this question (as cited). Interestingly a major point in our argumentation in e.g. Trauth et al. (2018) is that in this decade-long debate the discussion often lies purely in semantics and depends very much on categories of perspective what "abrupt" means.

Consequently, different understandings and definitions of this term may exist, one being that in the context of climate transitions the change from (for example) wet to dry is faster than forcing. And indeed, as the reviewer says, the duration of the transition, regardless of whether it is 1,500 yr long determined using the nonlinear fit in this paper or whether it is ~885 yr long using the change point analysis in Trauth et al. (2018), is significantly shorter than the quasi-linear decline in insolation forcing. From the perspective of humans living in the area the description or categorization of this transition is not important. For human groups, it was much more relevant how quickly the environment has changed compared to the time they had lived. We highlight that aspect of dimension from human perspective, as this is relevant for the story of this paper after L195-196.

When it comes to the effect of this tipping point style transition, we even go a little further if we consider the rather long change from wet to dry: the short extreme events during the transition were probably much more significant to transforming human living environments than the comparatively slow change from wet to dry.

Reviewer: What is the mechanistic relation between the fluctuations and the termination? The fluctuations occur over a time period of 2.5 ka and it is unclear why they occur before, during and after the decline of the AHP. In some places you say the fluctuation are precursors to the termination, but they occur throughout. Currently, the paper reads as an interesting observation, but does not provide a mechanism or links the short-term fluctuations with a better understanding of the termination. This point needs to be clearer.

Authors: Yes, good point. There is indeed a mechanical relationship between the transition and the flickering. Compare our comments on the editor's question, the answer to the corresponding question from Reviewer #1 and the response to you (Reviewer #2) on the comment in L71–79. To make this connection however clearer to the reader, we have now revised the last two sections of the main text (L62–79 of the original version), without repeating the work of the authors cited though (e.g. Pausata et al., 2020, Scheffer et al., 2009, Dakos et al., 2013, and Hopcroft and Valdes, 2021). We have to emphasize that the focus of our paper concentrates on possible experimental evidence of the transition with flickering/possible early warning signals, but not on the mechanism itself, as this is provided by our empirical data. We hope

that the reviewer will understand this and is referred to the highly recommended literature of the colleagues that is cited in this context.

Reviewer: The main hypothesis of this paper is that the small fluctuations are real climate signals. Why are they not noise? Even the pronounced D-O events in GRIP were not believed until the publication of GISP2 that showed the same patterns (Grootes et al., 1993, Nature). I suggest removing the long-term average and looking at a time series of the variance in different size averaging windows. Is the variance between 7-5 ka statistically larger than the average noise throughout the Chew Bahir record? Then it would be worth checking whether the size of the variance is a function of sedimentation rate, i.e. does higher degrees of noise occur in higher sed. rate portions of the core? I think this is clear when comparing the large variance between 0-10 ka and 35-45 ka when sed. rate is high with 35-10 ka when sed. rate is low and variance is low as well (from Foerster et al., 2014). I'm not sure the method I suggested is the way to go, but for this paper to be published in NC, I think you must find a way to convince that you are not over interpreting noise.

Authors: We fully agree with the reviewer, the two essential questions are: (1) is the observed flickering real and (2) what is the possible mechanism (see above). Of course, we tried a lot more statistical methods, including the usual linear methods for investigating tipping points and their surroundings, recommended by the cited literature. In the end, however, we limited ourselves to the two most promising and comprehensive methods of detection in order not to turn the paper into a book: (1) a detailed look at the sediment, its coloration and composition, and (2) a solid analysis using recurrence plots and recurrence quantification analysis. In our opinion, both show very clearly that the observed fluctuations are not random and thus noise (see also the consideration of synthetic examples in Trauth et al., 2019).

In addition to the sedimentary evidence discussed in more detail in the supplement, we also consider the repeated occurrence of the same diagnostic pattern over the past ~620 kyr BP to be a strong argument that this is an important regional climate phenomenon – independent from the coring technique and XRF scanning technique. Unfortunately, the suggestion that the signal-to-noise ratio might depend on the sedimentation rate cannot be tested due to the low temporal resolution and uncertainties of the RRMarch2021 age model, but it is absolutely plausible. This could, in addition to actual climatic causes or reasons in the sedimentation process, explain why the described transitions in the past ~620 kyr are very similar, but not perfectly congruent.

Reviewer: The fluctuations that are at the center of this paper are interpreted as representing a climatic signal. However, I think that internal sediment dynamics could be an additional possible source of these fluctuations. I'm going to play the devil's advocate in the hopes that this will help make the climatic interpretation more robust.

Authors: We agree that a robust understanding of the climate proxy K is key to the interpretation of EWS and flickering in the Chew Bahir pattern. We appreciate the concern by Reviewer #2 to provide more of the essential details of our previous studies to then make climate-based inferences on EWS. The K signal in the Chew Bahir sediment cores and modern surface samples from across the basin and Chew Bahir catchment could be shown to be primarily controlled by the hydrochemistry of the lake and pore water (Foerster et al., 2018; Gebregiorgis et al., 2021; Arnold et al., 2021; McHenry et al., 2023), which is a common mechanism in saline and hyper alkaline eastern African lakes (e.g. Singer and Stoffers, 1980; Deocampo et al., 2009; Deocampo et al., 2017).

In a study specifically concentrating on wet-dry transitions in Chew Bahir with the herein discussed termination of the AHP, we were able to show that the strong authigenic component present in the presented data of the corresponding mineral assemblages (XRD of whole-rock samples and clay separates) of the Chew Bahir sediments along the K record, suggests that the changes in K abundance are associated with both octahedral Mg-enrichment and low-temperature illitization (Foerster et al., 2018) and co-occurs with analcime abundances, all features of authigenic mineral formation. The formation of trioctahedral phases can be directly attributed to the authigenic alteration of minerals that is controlled by

the sensitive reaction of clay minerals with the highly saline and alkaline brines (McHenry et al., 2023; Foerster et al., 2018).

In this context, we would like to refer again to the detailed discussion of the K signal in connection with the indications in the sediment, especially in connection with the photographs of the core in Suppl. Fig. 2. Here it becomes clear that the extreme K events indeed correspond to different depositional milieus recorded in the sediment and are not simply artifacts of the internal core structure such as cracks or holes.

Reviewer: An alternative source of these fluctuations is that as the lake is drying out, older lacustrine sediments become exposed at the flanks of the lake and get washed into the lake and accumulate at the margin. Mixing of early Holocene sediments (high K+) with the new ~5.5 ka formed sediment (low K+) is a potential, non-climatic source of this flickering. The fluctuations appear most prominently in the core from the margin of the lake and not in those from the center of the lake. The authors explain this observation by an enhanced sedimentation rate at the margin. However, from Foerster et al., 2014 it seems that CB01, CB02 and CB06 have the same sedimentation rate. So, if the fluctuations are climatically driven, they should have shown up in both the margin core and the center of the lake core, but they show up only in the margin core, which might strengthen a sedimentary mixing origin of these fluctuations.

Authors: We acknowledge that considering all potential basin dynamics that could influence the K signal is crucial for a reliable proxy interpretation, and that the input of K via physical weathering and erosion from the slopes (or re-mobilization as suggested by Reviewer #2) of the sparsely vegetated rift shoulders through strong rain events (Foerster et al., 2012; Foerster et al., 2014; Arnold et al., 2021) could play an important role in detrital source-to-sink sediment dynamics. However, the physical weathering and erosion cannot explain the sensitive adjustment of the crystal chemistry of authigenic minerals evident in the sediment, including the progress of K-fixation in clay minerals that is evident in the Chew Bahir deposits, being of post-sedimentary origin. This includes the well-studied interval along the AHP termination (Foerster et al., 2018; Gebregiorgis et al., 2021). The reviewers' comment reflects the question of identification of detrital versus authigenic phases, a decade-old matter of debate (Jones, 1986; Larsen, 2008; Deocampo, 2015), and a concern we were able to address with detailed analyses in Foerster et al. (2018), where we showed that the major driver of K abundances must be of authigenic origin, though a detrital component can never be ruled out. Moreover, the absolute amount of older lacustrine sediments mixed with the younger ones by the process of fluvial erosion might be relatively low, regarding the huge catchment area even behind each small creek and the topography of steep slopes in the near neighborhood. Therefore, each heavy rainfall event would have activated alluvial fans in the surrounding which are mainly built of erosional sediments from the catchment and this material dominated the input of sediments via fluvial erosion into the Chew Bahir basin.

Second, aiming at a better understanding of the role of (modern) sedimentary processes in chemical proxy formation in Chew Bahir, a study on modern sedimentation and authigenic mineral formation used new mineralogical and geochemical data from modern sediments in the Chew Bahir basin and catchment (Gebregiorgis et al., 2021). This study was able to look in detail into the sedimentary processes in the Chew Bahir catchment, using oxygen isotopes in authigenic clay minerals, geochemical modelling and combining results from grain size analyses, whole-rock and clay mineralogy and X-ray core scanning. Third, the typical presence of authigenic products such as Mg-rich authigenic clay minerals, low-temperature authigenic illite and analcime in Chew Bahir sediments during evaporative phases could be confirmed by an approach using advanced hyperspectral analysis in the spectral range from 0.25 to 17 μm (Arnold et al., 2021). Samples along the Chew Bahir long cores (~620 ka) reflect the same pattern that is evident in the XRD and XRF records and show that authigenic Mg-enriched clays in the indicative absorption band ratio 2.3/2.2 μm (Mg-enriched clays) correspond with peaks in absorption bands 1.16 μm (analcime) and K abundances during the last ~620 ka BP (Arnold et al., 2021). These findings are in line with the efforts to use authigenic minerals as sensitive climate proxies (McHenry et al., 2021).

However, to consider the hints of the reviewer, we have revised the section starting with "... " in the chapter "The Chew Bahir record ..." We hope that the relationships between K concentrations, hydrochemistry and

climate are now clearer than they were in the original version of the manuscript. For more in-depth explanations, however, reference must be made to the literature cited, especially the work from our group mentioned above, in order not to allow the manuscript to grow any further (see also comment from Reviewer #3, "... what was new in this paper compared to previous papers published by the authors."). On the topic of sedimentation rates of the short cores CB01, CB02 and CB06, please also see our response to this reviewer's comment on L227 below.

Reviewer: In addition, if the fluctuations formed due to sedimentary mixing, then the ^{14}C ages throughout this time interval should also be mixed. The sed. rate for CB01 in Foerster et al., 2014 shows that the sediment rate is not linear during this time interval and that there are potentially large changes in the sedimentation rate (a rapid sedimentary rate increase around 7ka and then a drop in sedimentation rate after that). These are smoothed out when using the linear model compilation. This rapid change in sedimentary rate could also be an important clue strengthening the sedimentary mixing origin of the signal. This alternative mechanism should be explored, and the two observations need to be addressed before accepting the interpretation of the flickering as having a climatic origin.

Authors: Please refer to the discussion on sedimentation rates above. Our age model is a model that can only inadequately reflect the true fluctuations in sedimentation rates – as very common in models per se. Abrupt changes in sedimentation rate in the age model of CB01 should not be used to explain any phenomenon in the time series. To exclude an influence of the age model on the interpretation of the K curve, we now included a plot of the K content over depth in Suppl. Fig. 2.

Reviewer: L31. The abstract boils down to a local phenomenon and it is unclear why this flickering is of importance to a wider audience. Are there mechanistic reasons for this flickering? Do you expect this to occur in other regions? etc.

Authors: See our reply to Reviewer #1 on the same topic. Also see response to this reviewer on his comment on L71-79. Please also note that, as said in the paper, the flickering has been predicted to occur elsewhere by the models, and (as said in the paper) there are indications that it can also be seen in other records of African climate, even if it was initially overlooked.

Reviewer: L71-79. This seems like a very interesting point, but throughout the whole paragraph is unclear what are the possible types of tipping point for the AHP. The two options in lines 69-70 are unclear in respect to the AHP termination. I suggest elaborating and maybe presenting hypothesis for what you would expect to see for different types of tipping points. You mention the tipping point models in passing a few times in the paper, so I think it's worth to say a little bit more about what model these are and what conclusions you draw from them.

Authors: We understand the reviewer's point and have completely revised the last two paragraphs accordingly, starting with "Recent literature...". It now provides an overview of the two basic types of tipping points as well as the underlying mechanisms, as far as is known to current science. The stochastic noise in the climate system plays a particularly important role in relation to the main mechanism, the orbitally controlled decline in solar radiation. The first type of tipping between two stable states is characterized by a critical slowing down and a decreasing recovery from perturbations near the transition. Because of the generally high noise level in the African Monsoon System, it tends to exhibit the second type, transition with flickering.

Reviewer: L76-77. The model results of Hopcroft and Valdes 2021 show a large drying events at 7.7 ka which they claim could be a precursor to the collapse. This is very different form a series of small fluctuations you see in your record. So, I'm not sure "predicted by one recent modeling study" is an accurate statement.

Authors: We are pleased that the reviewer noticed this interesting drought event in the modeling results of Hopcroft and Valdes (2021), because in fact we also find a very similar event in our data at 7.7 kyr BP (Fig. 2A, marked by "7.7 ka drought" and L744 of the original manuscript). We discussed this event at

length with co-author Peter Hopcroft. In duplicate simulations (albeit with slightly different forcing) we didn't find an identical evolution of drying events. For this reason, we interpret this particular 7.7 kyr BP event as a coincidence in terms of the timing but model-data agreement when considering the overall characteristics of the flickering. However, we will keep an eye on it, especially since a similar (or even a similar pair, together with a less dramatic drought about 300 yrs prior to this event) of dry events also occurs at the transition around 379 kyr BP (Fig. 2B).

Apart from these thoughts, however, we would like to point out that the potentially irritating sentence in L76-77 no longer appears in the text and have carefully revised the last two paragraphs of the section accordingly (see our response to this reviewer's comment on L71-79).

Reviewer: L91. I suggest comma after Turkana basin and full stop after catchment.

Authors: This would mean that the Turkana Basin is the terminal sink for the sediment. To avoid unnecessary irritation, we have now deleted the second sentence as both the Neogene age and the maximum sediment thickness of five kilometers are speculative, the latter based on unpublished industrial gravity data in a repository at the Petroleum Operations Department in Ethiopia. Both, the age and the 2.5–5 km(?) sediment thickness are not relevant for the analysis of the ~300 m sediment cores.

Reviewer: L108-109. This line is basically the bottom line of the whole paper and I'm having a lot of trouble with it. The low K+ events occur before, during and after the termination, so I'm not sure why you claim that they are a precursor. Maybe they indicate that: 1) the termination wasn't so abrupt and 2) that the transition is characterized by enhanced variability. In addition, it seems like you are using the term "tipping point" as a specific point ("imminent") in time, but the record shows a 1500 year decline.

Authors: We are not sure whether we understand the reviewer's point. What is essential for an early warning signal is that it occurs *before* the actual event that it is intended to "warn" about. And since, as the reviewer rightly notes, the flickering occurs before the transition, it can serve as an early warning signal. In this respect, it is irrelevant whether the flickering also occurs during and after the transition. The duration of the transition, as already discussed in the reply above, is irrelevant when evaluating possible early warning signals, nor does it play a role in describing the transition as a tipping point, as long as it is faster than the corresponding forcing.

Reviewer: L111-112. What do these models show? It would be useful to add one sentence to explain, otherwise this is a pretty obscure sentence.

Authors: As already stated elsewhere, flickering has so far been shown in a single model experiment, namely that of Hopcroft and Valdes (2021), which we cite here.

Reviewer: L115. What MIS is this?

Authors: Assuming that the Reviewer #2 is referring to "marine isotope stage", we decidedly refrain here from using this time marker concept for our study as these marine isotope stages are derived from oxygen isotope ratios of benthic foraminifera, which reflect an indicator of global ice volume, and therefore the alternation of glacials and interglacials. Although very commonly used in the community to refer to specific time intervals we are very careful not to apply this concept to the CHB site in the tropics, one major reason being that the MIS in connection with wet-dry fluctuations in the Chew Bahir Basin might reflect an obsolete interpretation of glacial-interglacial forcing of low latitude climate change. Regardless of this, the exact timing of the transitions in the ~620 kyr record from the Chew Bahir Basin is neither relevant for the present discussion of possible tipping points nor possible within the uncertainties of the RRMarch2021 age model (Roberts et al., 2021). We therefore refrain from using the MIS concept for our record.

Reviewer: L123. In the methods section (L237) and caption figure line 618 the duration of CB01 is 9-4 ka, which is inconsistent with this sentence.

Authors: We thank this reviewer for pointing this potential point of irritation out to us and revised the caption accordingly. There is nothing wrong in L123, describing the width/height of blocks in RPs. Instead, the caption of Fig. 2 was wrong. We now corrected the caption, which now reads, "... between (A) 9–3 kyr BP and (B) 382–376 kyr BP interval ...", in agreement with what is shown in the figures.

Reviewer: L125. Do you interpret this as the tipping point occurring at 5.5 ka? you should say this explicitly and refer to the supp material.

Authors: This reviewer's comment addresses a very relevant question: where is the tipping point in the curve? It's probably not at 5.5 kyr BP; Instead, the point at ~5.5 kyr BP would be more likely the inflection point of the transition, not the beginning of the tipping. We had actually thought about it and discussed it with specialists in the field. In the end, we concluded that it was difficult to determine the beginning of the (physical, mechanical) tipping in the data, especially with high noise levels. Instead, one could describe the transition geometrically (and discuss it if desired) with a beginning, a change point, and an end. This is exactly what is attempted in the section "Determining the duration and start of tipping ..." in the supplement to which we now refer to in the corresponding main text. For a detailed discussion of the topic, also see the work by Livina et al. (2011, 2012), cited in the paper.

Reviewer: L160. I assume you mean the termination of the Holocene AHP?

Authors: Yes. We added "in the mid-Holocene" to clarify this point.

Reviewer: L166. What modeling results are presented here?

Authors: We added "HadCM3" to this sentence, which now reads "... the ~620 kyr Chew Bahir record, combined with the HadCM3 modeling results presented here ...".

Reviewer: L192. I think there is a contradiction here. You say (L189) that at 7 ka the site were abandoned, but here you say that the flickering starting at 6.5 ka is what caused the abandonment. Please explain.

Authors: Thanks for the comment, but that's not entirely correct. In fact, we write that the sites were abandoned at ~7 kyr BP, the flickering starts at ~6.5 kyr BP – so there is an approximate sign in front of the numbers. We don't want to conflate the two events when we don't know for sure? In our opinion, the agreement is good enough to suspect a connection, always considering that correlation does not necessarily mean causation.

Reviewer: L227. From the Foerster et al., 2014 it seems that CB01, CB02 and CB06 have the same sedimentation rate. This data seems contradictory to this statement. Please explain.

Authors: The reviewer refers to Foerster et al. (2014), which was submitted to *Climate of the Past*, but not accepted for publication. According to the policy of this journal, this text, together with critical reviews, remained publicly available with the value of an extended abstract in the sister journal *Climate of the Past Discussions*. All attempts to publish the article in another journal failed—among other things—because reviewers found the original paper and were inspired by the critical reviews, although the work was subject to major revisions since Foerster et al. (2014).

After almost four years of unsuccessful attempts to publish the work, we have abandoned the manuscript. The data from the short cores CB01–06, their correlation and age models, however, were used in another paper, namely Trauth et al. (2018) on change points during the past ~45 kyr BP, which is cited several times in this section on Chew Bahir setting, materials, and climate proxy. This paper contains a much more detailed statistical analysis of short cores CB01–06, including correlation using dynamic time warping and a discussion of the age models, than it was presented in Foerster et al. (2014). On the point of sedimentation rates, the 1st paragraph of the Discussion section of Trauth et al. (2018) reads:

"The CB01–03 cores were collected at increasing distances from the Hammar Range, which is the main sediment source, while the CB04–06 cores were collected towards the centre of the basin. The centre of the Chew Bahir basin is influenced to some extent by the Segen River, whereas the more western sites are increasingly influenced by runoff from the Hammar Range via the extensive alluvial fans, and episodically by the Weyto River. As a result there are significant differences in the type of sediment, the rate of sedimentation and, most importantly, in the potassium concentrations in the sediments as well as the variability of these potassium concentrations. These differences can be used to separate influences that are specific to particular coring locations from regional influences with features that are clearly common to all areas. The AHP and the mostly dry period during the YD stadial can, for example, be identified in all of the investigated cores and the dry-wet-dry alternations during the D-O events in most of them, while some of the less distinct dry-wet transitions during glacial times can only be observed in the CB01, CB04 and CB06 cores, and to a lesser extent also in CB03."

However, for clarification, we have now included the reference Trauth et al. (2018) which is also added at the end of this sentence in line L227 to which the reviewer refers.

Reviewer: L247. Where do you compare the record to the model? I didn't find this in the paper. Do you mean on line 111-112? If so, this is very unclear.

Authors: Assuming that the reviewer means that we compare the K record with the modeled precipitation data, we do so in L135–146 of the original version of the paper, starting with "Examining the RP/RQA results of the modeled precipitation reveals very similar structures despite the differences due to the different character of the climate variable."

Reviewer: L247-255. The Hopcroft and Valdes 2021 model was run using three different model configurations (standard, enhanced convection and dynamic vegetation). I assume you used the standard? Could you say a word on why you chose this run as opposed to the other two options?

Authors: In the methods of the submitted text we state that we use the "palaeo-conditioned" model which refers to the configuration in Hopcroft and Valdes (2021) with updates to both convection and dynamic vegetation. The other *three* configurations (standard, tuned convection, tuned vegetation) do not show good representations of the mid-Holocene state against which they were conditioned (or tuned). Specifically *standard* and *tuned convection* both show no greening of the Sahara under mid-Holocene conditions which is similar to the majority of other coupled climate models (e.g. Brierley et al., 2020). Also *tuned vegetation* shows only a moderate vegetation advance. The *tuned convection+vegetation* simulation shows a convincing greening which when run in transient mode also shows excellent agreement with records from the NW Africa against which it was not tuned (especially the precipitation results from Tierney et al., 2017).

We have clarified this in the revised text.

Reviewer: L284. See comment on line 123.

Authors: See response on L123. The "analysis of the 9–3 kyr BP" is correct, as shown in Fig. 2.

Reviewer #3

Reviewer: A really interesting and well-written manuscript. The end of the African Humid Period is a classic example of a climate tipping point which is well-known within the palaeo community. The manuscript explores the potential for identifying 'early warning signals' of this abrupt change. I struggled initially to work out what was new in this paper compared to previous papers published by the authors. The Itrax data and chronology have all been previously published for the short cores (CB01-06) but the data for the long cores is new. The recurrence quantification analysis has also been published for the short cores, which possibly explains the lack of detail in explaining the method. I found I had to read Trauth et al. (2019), which provides some excellent figures of how the recurrence quantification analysis works. I do think that some more explanation of the recurrence quantification analysis is needed in this paper – or at the very least it is essential to reference the Trauth et al. (2019) paper earlier in the manuscript. At the moment it is only referenced in the Methods; it needs to be added somewhere in the section from line 117. Altogether, there needs to be a bit more transparency in what has been previously published, and what is new here. There is certainly enough to warrant a separate paper, but details from previous work either needs to be explained again, or need to be clearly signposted to find that information.

Authors: We would like to thank the reviewer for his encouraging words about our work! We note the reviewer's suggestion to explain recurrence plots earlier and cite Trauth et al. (2019), which provides a very detailed but easy-to-understand explanation of RP/RQA. In fact, we had great difficulty structuring the manuscript in the current standard format of Nature family articles, where a two-part structure (main text and appendix) has been replaced by a three-part structure (main text, methods and appendix). Some of the comments from the other reviewers also show that we were only partially successful in structuring the article and we tried to provide better overview and structure in the revised version.

The explanation of the recurrence plots is a good example, as they are mentioned for the first time (but only very briefly) in L118 of the main text of the original version, with a very brief explanation in one single sentence in L120–122:

"RPs are graphical displays of recurring states of a system, calculated from the distance between all pairs of observations, (if required) within a cutoff limit; RQA uses measures of complexity for a quantitative evaluation of the RP's small scale structures."

Previously, only the detailed paper by Marwan et al. (2007) was cited here; we have now added Trauth et al. (2019) to accommodate the reviewer's request to cite this paper earlier in the manuscript. We also add one more sentence explaining RR and DET, and now the short but complete introduction to RP/RQA now reads:

RPs are graphical displays of recurring states of a system, calculated from the distance between all pairs of observations, (if required) within a cutoff limit; RQA uses measures of complexity for a quantitative evaluation of the RP's small scale structures (Marwan2007, Trauth2019). Among these, the recurrence rate (RR), indicated by the density of black dots in the RP, describes the propensity of the system to recur in a particular time period (Marwan2007, Trauth2019, Trauth2021). The ratio of the recurrence points that form diagonal structures (of a minimum length) is a measure for determinism (DET) of the system (Marwan2007, Trauth2019, Trauth2021).

A more detailed explanation of RP/RQA can then be found in the Methods section, beginning in L274 of the original version of the paper:

"Recurrence plots (RPs) are graphical displays of recurring states of a system, calculated from the (e.g., Euclidean) distance between all pairs of observations, (if required) within a cutoff limit (Marwan2007, Eckmann1987, Trauth2019, Trauth2021). The visual inspection of RPs is often complemented by a recurrence quantification analysis (RQA), which uses measures of complexity

for a quantitative evaluation of the RP's small scale structures (Marwan2007, Trauth2019, Trauth2021). Among these, the recurrence rate (RR) is measuring the density of black dots in the RP, describing the propensity of the system to recur in a particular time period (Marwan2007, Trauth2019, Trauth2021). Diagonal lines in RPs are diagnostic of predictable behavior in time series and can be used to predict future conditions from the present and past. The ratio of the recurrence points that form diagonal structures (of a minimum length) to all recurrence points is a measure for determinism (DET) of the system (Marwan2007, Trauth2019, Trauth2021)."

Here we cite Trauth et al. (2019) and the two overview papers about RP/RQA by Marwan et al. (2007, 2008) including an explanation of the RQA measures RR and DET and followed by the choice of the RP settings such as m , τ and w . We think that this will be sufficient for most readers; those readers with a greater interest in the RP/RQA topic will find it in the cited literature, in all cases with extensively commented syntactic examples and MATLAB codes.

The desire for more detail on RP/RQA surprises us somewhat because the reviewer, on the other hand, notes that not everything shown is new. Of course, this paper is one in a long series of 20+ publications on the Chew Bahir project, as not unusual for projects of this scale. In fact, we noticed the recurring droughts in the short cores several years ago and suspected that they were related to the tipping point, but the focus of those earlier papers was completely different. For example, in Foerster et al., 2015 and Trauth et al., 2015 we compared the AHP termination including those drought events to the archaeological record in nearby potential refuge areas. About 10 years ago, when many of our studies on the Chew Bahir short cores evolved, was at a time when the (cited) literature and related research on precursor events was still at an early stage and developing, which we followed with great interest. The breakthrough in our project in terms of EWS and tipping points came (and this is actually new) when we identified very similar transitions and precursor events several times in the long core as well. Up to this point, we could not be sure (and agree with Reviewer #1) whether these 20–80 year droughts were real. However, after identifying those patterns of diagnostic drought events several times in the long core, obtained with a different drilling technique and measured with a different XRF scanner in a different laboratory, the data provided evidence that the identified pattern in the mid-Holocene could not be a coincidence.

Reviewer: The manuscript mentions that the Chew Bahir record shows many of these transitions from dry to wet. It is not clear how many of these show this 'flickering' behaviour. Only two time periods appear to have been analysed – the Holocene and 382-376 kyr BP. Why only these two? It would be interesting to add some (brief) discussion regarding how many of the transitions back in time show the flickering behaviour, and for those that don't, why don't they? Is it a data resolution/proxy problem, or is flickering not apparent? I am also not sure how the wet-dry transitions highlighted in Suppl. Fig. 3 were chosen (A-K). There are several other periods of time that show an abrupt change in potassium, but are not highlighted. The figure caption states that the 10 examples 'differ in the occurrence of flickering and early warning signals', but I would like some more detail on this.

Authors: We would like to thank the reviewer for his question, as it gives us the opportunity to explain how we selected the examples of transitions presented in the paper. As indicated in line L732 of the original version of the manuscript, we find a large number of transitions of different structures in the ~620 kyr long record, something that would be expected. Some of those transitions are in fact very similar. From this cohort we have simply selected the 10 best examples – and that is why we show them in Suppl. Fig. 3. The paper comprises the full data set and associated MATLAB scripts, so the readers are invited to browse the time series themselves to study these 10 transitions and of course in addition all the other examples. In L740–741 we add, that these transitions all show flickering, but with varying degrees of clarity.

The transition between 380.2–377.2 kyr BP is the one which shows the highest similarity to the termination of the AHP (L741). That's why we describe this transition in greater detail in the following lines of the paragraph, but also point out the similarities and difference in the duration and shape of the other 8 transitions shown in Suppl. Fig. 3 after L753. The very detailed sedimentological and statistical

investigation of more, but similar transitions add more text and figures to the paper, without actually gaining additional insights, and therefore we have limited ourselves to this selection of 10 or 2 clear examples of transitions, respectively.

Reviewer: Given the journal's broad audience I would consider whether all the acronyms are needed, and whether some can be spelled out. The section from Line 117 is very heavy on the acronyms, which makes it difficult to follow. I don't think DET is ever spelled out – presumably this is Determinism.

Authors: The reviewer is absolutely right and we now spell out these acronyms in this paragraph. Both RR and DET are also spelled out in the Methods section again (L274–283) of the original version of the manuscript. In general, we replaced "EWS" by "early warning signals" in order to reduce the number of acronyms in the text.

Reviewer: I am surprised that there are only 2 figures, but they are highly detailed. Figure 1 from Trauth et al. 2019 contains a much better map for panel A – in this figure it is difficult to tell where the site is located in relation to the plateau. Figure 2 would probably benefit from having the y-axis labelled to help to decipher the figure for anyone not familiar with recurrence quantification analysis.

Authors: We cannot follow the reviewer here. The presented work contains nine figures, most of which have several panels and provide detail that could be separated into several separate figures, which is however constrained by the space and length of the journal so that the main text comprises two figures, whereas the 7 more figures are in the supplement. Figure 1a shows the location of the Chew Bahir Basin very clearly, while in Figure 1b the location of all short cores CB01-06, the intermediate length core CHB14-1 and the twin long cores CHB14-2 are located in the basin. Compared to a similar geological map in Trauth et al. (2019), this map was completely recreated using the base maps cited in the caption. In contrast to the old map, this new one breaks down the geological units and rock types in much more detail, which improves the interpretation of the sediments and the climate signals they contain, which is important and relevant for this work presented here.

Figure 2 was created with the same, but slightly improved MATLAB script as in Trauth et al. (2019, 2021). In Trauth et al. (2019), numerous synthetic examples of data and their RPs are also shown, which were also created with the same MATLAB script. This should enable readers to understand the RP and RQA analyses. Here, typical patterns in climate time series (e.g. noise, sinusoidal oscillations, different types of transitions and trends) are shown, their RPs are explained and the results of the RQA (e.g. RR and DET) are explained.

In contrast to the figures in Trauth et al. (2019, 2021), we have, however, added some markers in the three panels of the figures, namely arrows for trends, transitions and extreme events, as well as text labels. We feel that this is not a disadvantage of the figures, since it does not add too much detail. On the contrary, it is intended to help readers to follow our explanations in the text, even if they have not read the more specific literature on RP and RQA. It should be noted that *Nature Communications* is an online-only journal; there will be no printed version of the paper in which these figures would be difficult to read. Instead, readers can zoom in on the illustrations and easily see all the details.

We added a y-axis to Fig. 2 and Suppl. Figs. 4–6 the RPs as suggested by the reviewer – thanks for this suggestion!

Reviewer: Line 95: Sentence fragment.

Authors: Fixed. There was a superfluous "which" in the sentence here.

Reviewer: Line 107: I would reverse the sentence to first say 'nearing[?] the end of the AHP in the short cores from Chew Bahir, we observe.....' Makes it clearer on first read what time period is being discussed.

Authors: We thank this reviewer for this comment and reversed the sentence accordingly.

Reviewer: Line 108: strangely written... 'would have allowed a prediction of climate change'. By who? Or do you mean that we can use them to postdict? I'm not sure predication is necessarily the most helpful word here.

Authors: We absolutely understand the reviewer's point – but how should we put it differently? Of course, this is a fictitious, academic statement that one could have predicted the tipping of the climate if we had had the tools back then. So the answer to "by who" would be that a person seven thousand years ago, who recorded weather in Chew Bahir and calculated RPs/RQAs from the data collected, could have predicted the climate tipping. That's what EWS are for!

Reviewer: Line 110: This sentence talks about 'later' but it is not clear later from what time. I think there needs to be first some discussion of the nature of the transition – the idea of a tipping point suggests that it is step change, but the record shows that this transition did occur over a period of time (but faster than the forcing, which is the key point). The ramp fit analysis is good but it is never mentioned in the main manuscript.

Authors: What is meant by "later" is stated in the sentence itself, after ~6 kyr BP, since the sentence reads, "Later in the transition, after ~6 kyr BP, ...". The transition is not a step change, but rather a ramp-shaped transition, as described in the section "Determining the duration and start of tipping in the Chew Bahir record". The ramp is actually shown as a purple dashed line in Fig. 2A and B.

To add clarity, we now slightly modified the caption of Fig. 2, adding the information that the duration of the transition is determined by nonlinear least-squares fitting a ramp function (dotted purple line) to the K curve. We have also added that the extremes between which climate flickers during the transition are wet and dry, and that these extremes are marked by blue and red arrows in the figure, respectively.

Reviewer: Line 119: in the time series over what period of time? The whole record?

Authors: We modified this sentence to "... was analyzed section by section (within ~6 kyr windows) using methods from nonlinear dynamics ...".

Reviewer: Line 161: what do you mean by quasi-natural?

Authors: We modified this sentence to "... is quasi-natural (without anthropogenic overprints) because ...".

Reviewer: Line 165: this sentence seems to imply this paper is the first to show real world data that supports the hypothesis that there can be identifiable early warning signals of climate tipping points. Of course, there are many papers that have shown real world data to support this notion (some even on palaeoclimate data from other monsoon systems e.g. East Asian and West African monsoon). Please rephrase this sentence, or add appropriate references.

Authors: We modified this sentence to "... predicted only in theory for the region."

Reviewer: Line 190: You say at ~6.5 ka but then describe this as a period. If talking about a period of time please include a start and end date.

Authors: We modified this sentence to "About ~6.5 years ago and afterwards, i.e., the main flickering period ..."

Reviewer: Line 224-226: Why were some cores measured with Mo tube and others a Cr tube? Was it a deliberate choice or simply due to the location of the cores?

Authors: Correct, this was because the cores were located and measured in different labs (U Cologne vs. LLO), both with ITRAX scanners, but different tubes. Together with the numerous other measurement parameters to be set, such as measurement time and beam width, this changes the absolute number of K counts, but is irrelevant for the study and interpretation, due to the standardization of the data.

Reviewer: Line 222 down: It is quite confusing to work out what CB01-06 refers to. Perhaps you can add the words 'composite core' as otherwise it is necessary to look through the supplementary material to understand the core naming. Perhaps the word 'sequence' can be used for multiple cores in the same hole, and the word 'core' for a single drive.

Authors: We are not sure whether we got the reviewers concern here right. CB01–06 are six different short cores from different locations across the basin. We indicate that in the previous paragraph, list those cores in Suppl. Tab. 1 and hopefully most helpful, show this on the map in Fig. 1. CHB14-1 is another ~40 m long core from the center of the basin, while CHB14-2A and B are twin cores, stitched together to a composite core (CHB14-2). We agree these are many cores and added the reference to the corresponding Figure panel in Fig 1 for more clarity. We never mention multiple cores from the same hole, although (of course) each of the cores has multiple sections (as we call it) which are stitched together to the individual cores CB01–06, CHB14-1 and CHB14-2A and B.

Reviewer: Line 233: Please make clear that the age model is from ref. 63 but recalibrated using IntCal20 (as clarified in line 270),. It is a bit surprising that a linear age model is used when there are much more sophisticated Bayesian models widely available, but I acknowledge that this would be unlikely to affect the results.

Authors: We clearly say in L233 of the original manuscript that "The shorter cores CB01–06 were dated by the radiocarbon method, recalibrated using IntCal20", citing Reimer et al. (2020). The reviewer correctly assumes that we did indeed undertake extensive experiments in 2016 to improve the age model with the support of M. Blauuw (BACON), without success, however.

The problem with the short cores CB01–06 is that the team commissioned with the sampling at the time took samples (some of the total organic sediment) very indiscriminately and submitted them for dating (see Trauth et al., 2015, Fig. 2A). Especially in the younger area, there are numerous outliers, although these do not follow a uniform pattern and thus an elimination of outliers appears arbitrary. We have therefore opted for a very simple linear age model that supports the ¹⁴C data. This age model has a constant sedimentation rate, with the exception of the dry episode, which corresponds to the last glacial of the high latitudes and shows extremely low sedimentation rates across many sites in eastern Africa.

Reviewer: Line 234 and Line 242: Strange use of the word 'find'. 'Develop' an age model?

Authors: We agree and replaced "find" by "develop" in both L234 and 242.

Reviewer: Line 244: what datasets? The Itrax data?

Authors: Yes, the μ XRF data sets. We added " μ XRF" to the sentence.

Reviewer: Line 267: rephrase 'was subjected to'

Authors: Done, now it says, "... analyzed section by section (within ~6 kyr windows) using methods from nonlinear dynamics ...".

Reviewer: Line 270: how did you choose the 10yr interpolation window? What is the average timestep of the data? Was there missing data? Were there some periods where this may have overinterpolated the data?

Authors: We are not sure what the reviewer means by "10 yr interpolation window". The original measurements are evenly spaced along the core but become non-evenly spaced due to fluctuations in the sedimentation rate when converting the x-axis (in mm) to the t-axis (in yrs). One of the most important textbook rules when interpolating space series in time series is not to increase the number of data points. According to Trauth et al. (2019), CB01 with 2812 XRF measurements and a base age of 45,358 yrs BP, has a mean spacing of ~16 yrs, ranging from ~4 yrs in the upper part of the core to almost 2 kyrs in the deeper part of the core. We use the upper half (<20 kyr BP) of the core only; we interpolate the data to an evenly spaced time axis with 10 yrs intervals. This is explained in much greater detail in earlier papers (e.g. Trauth et al., 2015, 2018, 2019), which we cite in this new paper and would like to refrain from repeating all this information again in this paper.

Reviewer: Paragraph from line 274 (and corresponding line in main text): this is not sufficient detail. What is meant by pairs of measurements? What is the cut-off limit? There needs to be a clearer explanation of how this works. Similarly, the recurrence rate explanation is not clear. Over what time period does it measure the density of black dots? For each time step of 10 years?

Authors: We believe that all this, is explained in great detail with synthetic examples and MATLAB code in Trauth et al. (2019) and refer to the specified paper instead of repeating the details in full in the current paper.

Reviewer: Paragraph from line 296: Description of the change point/ramp fit analysis is well explained. But how it is used in the analysis is not clear. It is true that defining a particular point of tipping is not necessarily meaningful. But does the flickering occur before the tipping or is it part of the tipping? I.e., once a system starts to flicker, is the tipping locked in, or is it reversible? This seems to be the most interesting question.

Authors: See our explanations on the same topic above. We thank the reviewer for catching that we indeed forgot to explain the purple dashed line in Fig. 2! The purple dashed line was calculated by one of the three methods presented (i.e. ramp fit). From the explanations in the Supplement, as the reviewer very correctly recognizes and said in the text, it is clear that defining a specific point as the beginning of tilting is not possible, but also not useful. However, the flickering, as stated above and shown in Fig. 2, begins well before any of these points and can therefore be interpreted as a precursor signal. The flickering continues during the entire transition and beyond, as indicated by the model and described in the corresponding text. After the transition, also illustrated in Fig. 2, the flickering subsides and the climate "calms down" considerably to remain in the second stable state "dry".

References (not cited in the manuscript)

- Charney, J. et al. Drought in the Sahara: A biogeophysical feedback mechanism. *Science* 187, 434–435 (1975).
- Deocampo, D.M. Authigenic clays in lacustrine mudstones, in Egenhoff, S. et al., eds., *Paying attention to mudstones: Priceless!* Geological Society of America Special Paper 515: 49–64 (2015).
- Deocampo, D.M. et al. Saline lake diagenesis as revealed by coupled mineralogy and geochemistry of multiple ultrafine clay phases: Pliocene Olduvai Gorge, Tanzania. *American Journal of Science* 309 (9), 834–868 (2009).
- Foerster, V. et al. 46 000 years of alternating wet and dry phases on decadal to orbital timescales in the cradle of modern humans: the Chew Bahir project, southern Ethiopia. *Climate of the Past Discussions* 10, 977–1023 (2014).
- Jones, B.F. Clay mineral diagenesis in lacustrine sediments, in Mumpton, F., ed., *Studies in Diagenesis: U.S. Geological Survey Bulletin* 1578, 291–300 (1986)
- Larsen, D. Revisiting silicate authigenesis in the Pliocene-Pleistocene Lake Tecopa beds, southeastern California: Depositional and hydrological controls. *Geosphere* 4, 612– 639 (2008).

Singer, A., Stoffers, P. Clay mineral diagenesis in two East African lake sediments. *Clay Minerals* 15, 29–307 (1980).

REVIEWERS' COMMENTS

Reviewer #1 (Remarks to the Author):

The authors addressed my previous queries almost entirely. I particularly appreciate the detailed responses. Only two minor points remain that I believe the authors can easily and without much effort address prior to publication.

1.

Other data sets showing extreme excursions and/or flickering at the termination of the AHP include a lake record from Lake Dendi33, a diatomite record from Lake Abiyata34, and a record from Congo stalagmites35.

I still don't think this statement is entirely accurate, for reasons already mentioned in the first round of reviews. The datasets from Lake Abiyata and the Congo Stalagmites are too low in resolution. The elemental records from Lake Dendi suggest flickering during the African Humid Period (AHP) rather than during the transition out of the AHP. A Lake Dendi δD leaf wax record (Jaeschke et al. 2020) suggests a gradual, quasi-linear decline in precipitation between 8 and 2 kyr. As such, I suggest the following rewording of the respective sentence:

Other data sets showing excursions at the termination and flickering during the AHP include a lake record from Lake Dendi33, a diatomite record from Lake Abiyata34, and a record from Congo stalagmites35.

On a more minor note, "diatomite" is probably not the appropriate term here given that the sediments described in the respective Lake Abiyata paper are unconsolidated. The correct term for unconsolidated sediment with diatoms comprising more than 50% by weight is "diatomaceous ooze." However, the paper cited focuses 'only' on the diatom taxonomy/record. Therefore, I would suggest replacing "diatomite record" with "diatom record" or simply "lake record."

2.

I appreciate the authors' response to my previous query regarding the mechanistic understanding of the underlying causes of the flickering. I would suggest including a somewhat more expanded explanation of the previously published modeling study. See below:

L82: 'For the African Monsoon System, flickering prior to transition was recently predicted by a modeling study15,..'

Here I would expect the authors to insert a more detailed account on the actual driver of the transition. The authors provide a good example in the response to my previous query:

'In these studies, we found that the model displays strong positive feedbacks between vegetation cover and precipitation, in agreement with theoretical predictions (e.g. Charney et al., 1975). This is caused by both radiative and hydrological effects of vegetation. Firstly, the darker vegetation enhances solar energy absorbed fueling monsoonal type circulation onto land and hence precipitation. Vegetation as opposed to bare soil enables more efficient moisture-recycling back to the atmosphere, thus providing a pair of self-reinforcing feedbacks. In this configuration of the climate model, we find that these two effects are approximately equal in strength. The flickering arises as a result of the approach to a critical value of external forcing of the climate system. In this case the gradually declining summer insolation during the Holocene. As the threshold is approached, small perturbations induced by the simulated variability in the model can cause the model to flicker between states (Hopcroft and Valdes, 2021).'

A shortened version of the above, highlighting the vegetation-precipitation feedback, should suffice to

inform the reader appropriately.

Reviewer #2 (Remarks to the Author):

This is the second time I'm reviewing this paper. The current version is much better, specifically the rewritten paragraphs provide a much clearer view of the motivation behind the paper. I think the paper in its current version is almost ready for publication.

I do however still have two issues with the comparison with the climate model. Hopcroft and Valdes don't show the same type of flickering that you show in your record. They show one such event, which is much larger than what your record shows. In the paper you state that this model predicts the observations that you see (L. 83-84, 133), but this is not a precise statement. I suggest presenting a more nuanced version of these statements.

L. 159-161. This is the first time in the paper you say that you modeled precipitation. It would be useful to introduce the model and say what exactly you used - is the precipitation output above the lake? At what resolution? and refer to the methods, otherwise it's unclear.

Reviewer #3 (Remarks to the Author):

The authors have satisfactorily responded to my previous comments on the manuscript. There is more clarity in certain sections and some minor amendments to the figures which helps the interpretation.

Reviewer #1

Reviewer: The authors addressed my previous queries almost entirely. I particularly appreciate the detailed responses. Only two minor points remain that I believe the authors can easily and without much effort address prior to publication.

"Other data sets showing extreme excursions and/or flickering at the termination of the AHP include a lake record from Lake Dendi33, a diatomite record from Lake Abiyata34, and a record from Congo stalagmites35."

I still don't think this statement is entirely accurate, for reasons already mentioned in the first round of reviews. The datasets from Lake Abiyata and the Congo Stalagmites are too low in resolution. The elemental records from Lake Dendi suggest flickering during the African Humid Period (AHP) rather than during the transition out of the AHP. A Lake Dendi δD leaf wax record (Jaeschke et al. 2020) suggests a gradual, quasi-linear decline in precipitation between 8 and 2 kyr. As such, I suggest the following rewording of the respective sentence:

"Other data sets showing excursions at the termination and flickering during the AHP include a lake record from Lake Dendi33, a diatomite record from Lake Abiyata34, and a record from Congo stalagmites35."

On a more minor note, "diatomite" is probably not the appropriate term here given that the sediments described in the respective Lake Abiyata paper are unconsolidated. The correct term for unconsolidated sediment with diatoms comprising more than 50% by weight is "diatomaceous ooze." However, the paper cited focuses 'only' on the diatom taxonomy/record. Therefore, I would suggest replacing "diatomite record" with "diatom record" or simply "lake record."

Authors: We appreciate the reviewer's input, providing us with a complete sentence that we are happy to use exactly as it is, including "lake record" instead of "diatom record".

Reviewer: I appreciate the authors' response to my previous query regarding the mechanistic understanding of the underlying causes of the flickering. I would suggest including a somewhat more expanded explanation of the previously published modeling study. See below:

L82: 'For the African Monsoon System, flickering prior to transition was recently predicted by a modeling study15,..'

Here I would expect the authors to insert a more detailed account on the actual driver of the transition. The authors provide a good example in the response to my previous query:

'In these studies, we found that the model displays strong positive feedbacks between vegetation cover and precipitation, in agreement with theoretical predictions (e.g. Charney et al., 1975). This is caused by both radiative and hydrological effects of vegetation. Firstly, the darker vegetation enhances solar energy absorbed fueling monsoonal type circulation onto land and hence precipitation. Vegetation as opposed to bare soil enables more efficient moisture-recycling back to the atmosphere, thus providing a pair of self-reinforcing feedbacks. In this configuration of the climate model, we find that these two effects are approximately equal in strength. The flickering arises as a result of the approach to a critical value of external forcing of the climate system. In this case the gradually declining summer insolation during the Holocene. As the threshold is approached, small perturbations induced by the simulated variability in the model can cause the model to flicker between states (Hopcroft and Valdes, 2021).'

A shortened version of the above, highlighting the vegetation-precipitation feedback, should suffice to inform the reader appropriately.

Authors: We modified the text according to the suggestions of the reviewer, adding a slightly shortened version of the text from the response letter:

"In these studies, a strong positive feedback between vegetation cover and precipitation, caused by both radiative and hydrological effects. Firstly, the darker vegetation enhances solar energy absorbed fueling monsoonal type circulation onto land and hence precipitation, and secondly, vegetation as opposed to bare soil enables more efficient moisture-recycling back to the atmosphere, thus providing a pair of self-reinforcing feedbacks. In HadCM3 the albedo effect is stronger than the moisture effect by about a factor of three and the flickering arises as a result of the approach to a critical value of external forcing of the climate system, in this case the gradually declining summer insolation during the Holocene. As the threshold is approached, small perturbations induced by the simulated variability in the model can cause the model to flicker between states (Hopcroft2021)."

Reviewer #2

Reviewer: This is the second time I'm reviewing this paper. The current version is much better, specifically the rewritten paragraphs provide a much clearer view of the motivation behind the paper. I think the paper in its current version is almost ready for publication.

Authors: Thank you!

I do however still have two issues with the comparison with the climate model. Hopcroft and Valdes don't show the same type of flickering that you show in your record. They show one such event, which is much larger than what your record shows. In the paper you state that this model predicts the observations that you see (L. 83-84, 133), but this is not a precise statement. I suggest presenting a more nuanced version of these statements.

Authors: We can very well understand the reviewer's comment. Initially, it was Hopcroft and Valdes (2021) paper that attracted our attention and the idea for this paper in the first place. After a Hopcroft's visit to the University of Potsdam, we worked together on a way to compare the palaeodata (-> potassium concentration of the sediment as an aridity indicator in a lake basin in the southern Ethiopian rift) and the model data (-> modelled precipitation in the northern half of Africa, as said in the Methods section and in Hopcroft and Valdes, 2021). They are actually very different variables, but this is very common in palaeoclimate research: you can't measure precipitation directly in climate archives, you need a proxy, and there are many different proxies for precipitation (or what we think they are).

Recurrence plots or recurrence quantification analysis are a very good way of standardising the data, i.e. largely freeing it from the processes and variables, their dimension and physical unit. And here, but actually also in the time series itself, there are great similarities to the RPs of the palaeodata, not in Hopcroft and Valdes (2021), but in modeled precipitation as shown in our Supplementary Figure 6. Perhaps this figure was not correctly identified by the reviewer because the numbering of the Suppl. Figs. in the text was incorrect, Suppl. Figs. 4–7 were wrongly numbered as Figs. 6–9 in the text; we apologise for this mistake! In this figure, modeled precipitation shows a gradual but very instable (flickering) transition followed by the relatively stable dry conditions afterwards. However, our mistake, we accidentally reversed the y-axis of the time series plot, which is now fixed. We used the same script as for the paleo data in which high K is dry, and therefore the y-axis is reversed. Also the RPs look very similar, including the diagonal lines in the transition phase indicating that the flickering is actually cyclic.

We actually went back to the places in Lines 83–84 and 133 mentioned by the reviewer to see how we could improve the text, but didn't find a better solution. There are similarities, as said above, but we don't want to push the correlation further as they are different variables and different areas. Taking that into account, we think the similarities are quite obvious. We have revised Suppl. Figs. 4–6 again, removed the

grid in Suppl. Figs. 5+6 and added a colorbar, which now shows the Euclidean distance values of pairs of observations along the time series. We hope that this will make the similarities between the paleo time series and modeled time series and their corresponding recurrence plots more visible.

L. 159-161. This is the first time in the paper you say that you modeled precipitation. It would be useful to introduce the model and say what exactly you used - is the precipitation output above the lake? At what resolution? and refer to the methods, otherwise it's unclear.

Authors: As suggested by this reviewer, we modified the sentence in Line 159–161 to "Examining the RP/RQA results of the modeled precipitation from a transient climate model simulation of Holocene covering 10–0 kyr BP reveals very similar structures despite the differences due to the different character of the climate variable (i.e., sedimentary K concentrations versus simulated precipitation) (see "Methods" for more information about modeling)". In the Methods section, and also in the cited paper by Hopcroft and Valdes (2021), it is said that it is modeled precipitation of northern Africa, not precipitation over the lake. The methodology chapter also provides information on the spatial resolution of the model, "The horizontal resolution of the atmospheric model is 3.75° x 2.5° in longitude-latitude with 19 unequally spaced vertical levels. In the ocean the resolution is 1.25° x 1.25° with 20 unequally spaced vertical levels."

Reviewer #3

Reviewer: The authors have satisfactorily responded to my previous comments on the manuscript. There is more clarity in certain sections and some minor amendments to the figures which helps the interpretation.

Authors: Thank you!